

# Identifying and correcting interferences to PTR-ToF-MS measurements of isoprene and other urban volatile organic compounds

Matthew M. Coggon[1*], Chelsea E. Stockwell[1,2], Megan S. Claflin[3], Eva Y. Pfannerstill[4], Xu Lu[1,2], Jessica B. Gilman[1], Julia Marcantonio[5], Cong Cao[5], Kelvin Bates[1,2], Georgios I. Gkatzelis[6], Aaron Lamplugh[7], Erin F. Katz[4,8], Caleb Arata[4], Eric C. Apel[9], Rebecca S. Hornbook[9], Felix Piel[10,11,12], Francesca Majluf[3,a], Donald R. Blake[13], Armin Wisthaler[10,11] Manjula Canagaratna[3], Brian M. Lerner[3], Allen H. Goldstein[4,14], John E. Mak[5], Carsten Warneke[1]

[1]NOAA Chemical Sciences Laboratory, Boulder, CO, 80305, USA
[2]Cooperative Institute for Research in Environmental Sciences, University of Colorado, Boulder, CO, 80305, USA
[3]Aerodyne Research, Inc., Billerica, MA, 01821, USA
[4]Department of Environmental Science, Policy, & Management, University of California Berkeley, Berkeley, CA, 94720, USA
[5]School of Marine and Atmospheric Science, Stony Brook University, Stony Brook, NY 11794, USA
[6]IEK-8: Troposphere, Forschungszentrum Jülich GmbH, Jülich, Germany
[7]Institute of Behavioral Science, University of Colorado, Boulder, CO, 80305, USA
[8]Department of Chemistry, University of California Berkeley, Berkeley, CA, 94720, USA
[9]Atmospheric Chemistry Observations & Modeling Laboratory, NCAR, Boulder, CO, 80301, USA
[10]Department of Chemistry, University of Oslo, Oslo, Norway
[11]Institut für Ionenphysik und Angewandte Physik, Universität Innsbruck, Innsbruck, Austria
[12]IONICON Analytik GmbH, Innsbruck, Austria
[13]Department of Chemistry, University of California, Irvine, CA, 92697, USA
[14]Department of Civil and Environmental Engineering, University of California Berkeley, Berkeley, CA, 94720, USA
[a.]Now at Olin College of Engineering, Needham, MA, 02492

*Corresponding author: matthew.m.coggon@noaa.gov

**Abstract:** Proton-transfer-reaction time-of-flight mass spectrometry (PTR-ToF-MS) is a technique commonly used to measure ambient volatile organic compounds (VOCs) in urban, rural, and remote environments. PTR-ToF-MS is known to produce artifacts from ion fragmentation, which complicates the interpretation and quantification of key atmospheric VOCs. This study evaluates the extent to which fragmentation and other ionization processes impacts urban measurements of the PTR-ToF-MS ions typically assigned to isoprene (m/z 69, $C_5H_8H^+$), acetaldehyde (m/z 45, $CH_3CHO^+$), and benzene (m/z 79, $C_6H_6H^+$). Interferences from fragmentation are identified using gas-chromatography (GC) pre-separation and the impact of these interferences are quantified using ground-based and airborne measurements in a number of





US cities, including Las Vegas, Los Angeles, New York City, and Detroit. In urban regions with
low biogenic isoprene emissions (e.g., Las Vegas), fragmentation from higher carbon aldehydes
and cycloalkanes emitted from anthropogenic sources may contribute to m/z 69 by as much as
50% during the day, while the majority of the signal at m/z 69 is attributed to fragmentation during
the night. Interferences are a higher fraction of m/z 69 during airborne studies, which likely results
from differences in the reactivity between isoprene and the interfering species along with the
subsequent changes to the VOC mixture at higher altitudes. For other PTR masses, including m/z
45 and m/z 79, interferences are observed due to the fragmentation and secondary ionization of
VOCs typically used in solvents, which are becoming a more important source of anthropogenic
VOCs in urban areas. We present methods to correct these interferences, which provide better
agreement with GC measurements of isomer specific molecules. These observations show the
utility of deploying GC pre-separation for the interpretation PTR-ToF-MS spectra.
**1. Introduction**
Volatile organic compounds (VOCs) are an important contributor to urban air pollution. Once
emitted to the atmosphere, VOCs undergo chemical reactions that contribute to the formation of
hazardous pollutants such as ozone and secondary organic aerosol. It is important to quantify VOC
mixing ratios in urban areas to determine strategies that may reduce air pollution.
Proton-transfer-reaction time-of-flight mass spectrometry (PTR-ToF-MS) is a technique used to
measure a wide spectrum of VOCs, including oxygenates, aromatics, furanoids, nitriles, and
biogenic species such as isoprene and monoterpene isomers (Yuan et al., 2017). PTR-ToF-MS
measurements in urban regions enable the determination of VOC mixing ratios from an extensive
range of emission sources, including fossil fuels, solvent evaporation from volatile chemical
products (VCPs), residential wood burning, cooking, and urban foliage (Yuan et al., 2017). The
fast-time resolution and broad selectivity of PTR-ToF-MS measurements enables source
apportionment, flux estimates, and spatial mapping on mobile platforms that yield important
information about urban VOC source strengths (e.g., Gkatzelis et al., 2021a;Karl et al.,
2018;Pfannerstill et al., 2023)
VOC detection by PTR-ToF-MS relies on analyte reactions with protonated water (Reaction 1).
$$VOC \ + \ H_3O^+ \rightarrow VOC \cdot H^+ \ + \ H_2O \qquad (R1)$$
Proton transfer is exothermic and spontaneous for VOCs with a proton affinity that is greater than
water. For many VOCs, including ketones, aromatics, and nitriles, the protonated product
($VOC \cdot H^+$) is the primary signal detected by PTR-ToF-MS. For other VOCs, secondary reactions
including dehydration, fragmentation, and water clustering results in additional product ions that
can complicate the mass spectra. Pagonis et al. (2019) summarizes studies that have reported
fragmentation for a wide spectrum of species. Fragmentation is most prevalent in alcohols,
aldehydes, and other species with long-chain alkane functionality. Small alcohols and aldehydes
($C < 3$) primarily react to form protonated products following R1, while at higher carbon numbers,
a large fraction of the reactions follow dehydration and/or fragmentation (R2-R3).
$$(R - OH, RH = O) \cdot H^+ \rightarrow R^+ \ + \ H_2O \qquad (R2)$$



$$R^+ \rightarrow R_1^+ + R_2 \qquad (R3)$$
R is the carbon backbone of an alcohol (R-OH) or aldehyde (RH=O), $R^+$ is the dehydration product,
$R_1^+$ is a fragment, and $R_2$ is a neutral product. Fragmentation may also result from protonation of
cycloalkanes or alkyl aromatics. PTR-ToF-MS is not sensitive to small alkanes (C < 5), but larger
alkanes and cycloalkanes are detected at low sensitivity and upon ionization subsequently
fragment to produce ions that often overlap with the dehydration and fragmentation products of
alcohols and aldehydes (Arnold et al., 1998;Gueneron et al., 2015;Jobson et al., 2005). The degree
of dehydration and fragmentation is partially dependent on the strength of the drift field (E) and
density (N) (characterized by the E/N ratio), which impacts ion kinetic energy (Arnold et al.,
1998;Krechmer et al., 2018;Yuan et al., 2017;Holzinger et al., 2019). Lower E/N results in lower
fragmentation, but higher clustering with neutral water, which may further complicate the mass
spectra (Holzinger et al., 2019). Additional products may also be formed by reactions of analytes
with $O_2^+$ and $NO^+$ ions, which are present due to the ionization of small amounts of air in the
discharge region.
In the atmosphere, complex mixtures of emissions may result in PTR-ToF-MS mass spectra where
dehydration and fragmentation products interfere with the quantification of important atmospheric
VOCs. For example, PTR-ToF-MS measurements in regions with significant oil and natural gas
development show that substituted cycloalkanes fragment to produce significant signal at m/z 69
($C_5H_9^+$, Koss et al., 2017;Warneke et al., 2014;Pfannerstill et al., 2019). Likewise, interferences
have been observed downwind of urban and industrial environments (e.g., Inomata et al.,
2010;Choi et al., 2022). These fragments overlap with protonated isoprene and these previous
studies have shown that interferences make isoprene quantification challenging in these regions.
In forested areas, isoprene is largely emitted from biogenic sources and previous studies that have
compared PTR-ToF-MS measurements to those from gas-chromatography show good agreement
(e.g., Kaser et al., 2013). The impact of an interference to specific molecules, such as isoprene,
depends on atmospheric composition which changes spatially (e.g., urban vs. rural regions) and
temporally (e.g., summer vs. winter).
Assessments of interferences on PTR-MS measurements in urban atmospheres have been
conducted previously (e.g., Warneke et al., 2003), but the sources that contribute to urban VOCs
change on decadal timescales as fossil fuel emissions steadily decline (Kim et al., 2022;Warneke
et al., 2012). The urban atmospheric composition, once dominated by motor vehicle emissions, is
now composed of a higher proportion of oxygenates from solvents, other VOCs emitted from
sources such as VCPs, and cooking (Gkatzelis et al., 2021b;McDonald et al., 2018;Peng et al.,
2022;Wernis et al., 2022). Significant advances in PTR-ToF-MS detectors (quadrupole vs time-
of-flight mass spectrometers) and drift tube designs have enhanced instrument capabilities to
acquire mass spectra with greater resolution and sensitivity (Breitenlechner et al., 2017;Krechmer
et al., 2018;Yuan et al., 2016;Holzinger et al., 2019). These technological advances enable better
identification and quantification and also an improved understanding of the interferences that
impact PTR-ToF-MS spectra.
With the changes in atmospheric composition and technological advances, it is necessary to revisit
potential interferences to commonly observed and reported VOCs by PTR-ToF-MS. In this study,



we investigate the interferences that impact PTR-ToF-MS spectra measured across several US
urban areas including Los Angeles, CA, Las Vegas, NV, Detroit, MI, and New York City, NY.
Interferences are identified using GC pre-separation, similar to previous measurements that
quantified PTR-ToF-MS fragmentation and interferences observed in complex mixtures, including
wildfire and urban emissions (e.g., Koss et al., 2018;Warneke et al., 2003). We show that
commonly measured species, such as acetaldehyde, benzene, and isoprene, exhibit interferences
from larger molecules associated with solvent use and cooking. The extent of these interferences
depends on the temporal and spatial variability of VOC emission sources. We present methods to
correct for these interferences based on the measurement capabilities of modern PTR-ToF-MS
instruments.
**2. Methods**
Table 1 summarizes the key field campaigns and instrumentation used to quantify VOCs and PTR-
ToF-MS interferences. Multiple PTR-ToF-MS instruments are used in this study and Table 1
outlines the PTR-ToF-MS reactor designs and drift tube operating parameters that play an
important role in determining ion distributions. In this study, all instruments were operated with
E/N 120 – 140 Td. The instruments described in this study use ion-molecular reactors and ion
optics devolved by Ionicon Analytik and Tofwerk, AG as described by Müller et al. (2014) and
Krechmer et al. (2018), respectively. The following sections describe each campaign and provide
additional details of instrument operation
**Table 1:** Summary of campaigns, instrumentation, drift tube operating parameters, and interferences reported for each
campaign.

| Campaign | Location | Dates | Instrumentation | Reactor Design[1] | E/N (Td) | Studied Interferences |
|---|---|---|---|---|---|---|
| *SUNVEx* | Las Vegas, NV | July 1 – 30, 2021 | NOAA PTR-ToF-MS[2] NOAA GC-PTR-ToF-MS NOAA iWAS | Vocus | 140 | Isoprene, aromatics, oxygenates |
| *RECAP-CA* | Los Angeles, CA; Central Valley, CA | June 1 – Aug 30, 2021 | NOAA PTR-ToF-MS[2] UC Berkeley PTR-ToF-MS[2] NOAA GC-MS | Vocus (NOAA) Vocus (Berkeley) | 140 130 | Isoprene, aromatics, oxygenates |
| *FIREX-AQ* | Los Angeles, CA | September 5, 2019 | NOAA PTR-ToF-MS[2] Oslo PTR-ToF-MS[3] NOAA iWAS, UCI Irvine WAS NCAR TOGA-TOF | HC / DT (NOAA) HC/ DT (Oslo) | 125 120 | Isoprene |
| *MOOSE* | Detroit, MI | May 21 – June 30, 2021 | Aerodyne PTR-ToF-MS[2] Aerodyne GC-EI-ToF-MS | Vocus | 125 | Isoprene, aromatics |
| *LISTOS* | New York City, NY | January 2020 – April 2021. | Stony Brook PTR-ToF-MS[3] | HC / DT | 130 | Isoprene |

[1] HC / DT refers to the hollow cathode / drift tube design used in traditional PTR-MS instruments. This reactor is used in this study with both
Tofwerk and Ionicon systems. The Vocus reactor is used with Tofwerk instruments.
[2] Tofwerk design PTR-ToF-MS using quadrupole ion optics as described by Krechmer et al. (2018)
[3] Ionicon design PTR-ToF-MS with ion optics consisting of two einzel lens as described by Müller et al. (2014)
**2.1.Campaign Descriptions**
**2.1.1.  SUNVEx / RECAP-CA**
PTR-ToF-MS measurements were performed as part of the 2021 Southwest NOx and VOC
Experiment (SUNVEx, https://csl.noaa.gov/projects/sunvex/) and Re-Evaluating the Chemistry of



Air Pollutants in California (RECAP-CA). SUNVEx was a ground campaign conducted to study air quality in the Las Vegas Valley during the summer ozone season using both mobile and ground-based sampling. The RECAP-CA campaign was conducted in Los Angeles and included mobile, ground-based, and airborne sampling.

Measurements in Las Vegas were conducted between 30 June–27 July 2021 at an air quality monitoring station located near the Jerome Mack Middle School (Fig. 1). Jerome Mack is an urban background site located ~ 8km east of downtown Las Vegas. The site was chosen based on its suite of trace gas and $PM_{2.5}$ monitors and classification as a US Environmental Protection Agency (EPA) Photochemical Assessment Monitoring Station (Annual Monitoring Network Plan, 2022).

Measurements in the Los Angeles Basin were conducted between 2 August and 5 September 2021 at the California Institute of Technology in Pasadena, CA (Caltech, Fig. 1). The ground site was located within 0.5 km of the site used during the California Research at the Nexus of Air Quality and Climate Change (CalNex) field study in order to directly compare with air quality measurements conducted in 2010 (Ryerson et al., 2013). During this portion of the campaign, instruments were situated in a trailer and sampled air from the top of a 10-m tower.

During both SUNVEx and RECAP-CA, the NOAA mobile laboratory was deployed to sample the spatial distribution of VOCs and $NO_x$ in regions of varying population density. A similar sampling strategy was employed previously to study urban VOC enhancements in New York City and is useful for identifying VOC signatures emitted from major sources, such as fossil fuels, VCPs, and cooking activities (Coggon et al., 2021;Gkatzelis et al., 2021a;Gkatzelis et al., 2021b;Stockwell et al., 2021). Drive tracks from the mobile laboratory are shown on the map in Fig. 1, along with the locations of the ground sites and major population centers.



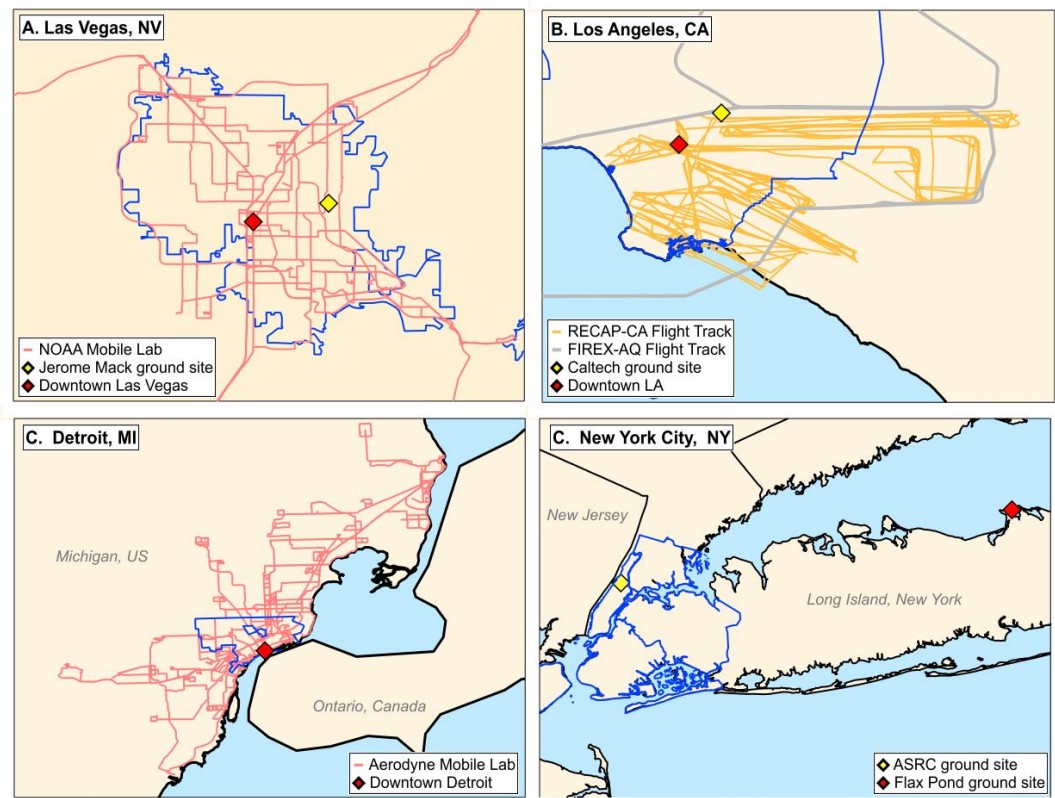

195

**Figure 1**. Mobile laboratory drive tracks, flight tracks, ground site locations, and locations of interest for the field campaigns outlined in Table 1. The blue lines highlight the statistical metropolitan areas for Las Vegas, Los Angeles, and Detroit, and the five boroughs of New York City.

The airborne component of RECAP-CA was conducted onboard the Naval Postgraduate School UV-18A Twin Otter aircraft and was based out of an airport located in Burbank, CA. Measurements of VOCs, $NO_x$, and greenhouse gases took place on nine days between 1 June and 23 June 2021. The Twin Otter typically flew at ~ 300 m above ground level at air speeds of 50 – 60 m s$^{-1}$. Each flight covered approximately 5000 km of distance across the Los Angeles area, including downtown, the coast, the Santa Ana area, and the San Bernardino Valley. The flight track of the Twin Otter is shown in Fig. 1b.

### 2.1.2. FIREX-AQ

The 2019 Fire Influence on Regional to Global Environments and Air Quality (FIREX-AQ) campaign was a large field study designed to investigate the emissions and atmospheric chemistry of biomass burning emissions. A detailed description of the campaign, instrumentation, and science goals is provided by Warneke et al. (2023). As part of the measurements, urban flights were performed through the Los Angeles Basin on 5 September 2019. VOC measurements were conducted onboard the NASA DC-8 by the NOAA PTR-ToF-MS and University of Oslo PTR-



ToF-MS. VOC measurements were also conducted by three GC instruments: the NOAA improved Whole Air Sampler (iWAS), the University of California, Irvine Whole Air Sampler (WAS) and the NCAR Trace Organic Gas Analyzer with a Time-of-Flight mass spectrometer (TOGA-TOF). The flight tracks conducted in Los Angeles are shown in Fig. 1b.

### 2.1.3. MOOSE Campaign

The Michigan-Ontario Ozone Source Experiment (MOOSE) campaign was a multi-institutional ground-based and mobile sampling effort conducted in 2021 to study ozone, meteorology, and pollution in and around Michigan and Ontario. This region is currently designated as non-attainment of the US federal ozone standard. Aerodyne Research, Inc. scientists deployed the Aerodyne Mobile Laboratory (Herndon et al., 2005;Yacovitch et al., 2015) as part of the CHEmical Source Signatures (CHESS) sub-experiment in order to measure emission plumes from point sources and gain insight to the drivers of local ozone pollution during MOOSE. Other goals of CHESS included developing emission source fingerprints for significant industrial source sites in the area.

Ambient VOC measurements were conducted onboard the mobile laboratory using a PTR-TOF-MS which was co-located with an *in situ* GC-EI-TOF-MS equipped with thermal desorption preconcentration (Claflin et al., 2021). The AML sampled air around the Detroit metropolitan region between 21 May and 30 June 2021 (Fig 1c). During mobile sampling, the mobile laboratory transited through major population centers and targeted industrial point sources. Overnight and when not driving, the mobile laboratory was stationed at the Salina Elementary/Intermediate Schools in Dearborn, MI, parked at the Michigan Department of Environment, Great Lakes, and Energy air monitoring station [AQS ID 26-163-0033].

### 2.1.4. LISTOS

The Stony Brook PTR-ToF-MS was deployed on the rooftop observatory at the Advanced Sciences Research Center (ASRC) of the City University of New York to make continuous, high time-resolution measurements of VOCs during the COVID lockdown from January 2020 to April 2021, (Fig. 1c; Cao et al., 2023). This campaign was a part of the broader Long Island Sound Tropospheric Ozone Study (LISTOS). ASRC is located in the Manhattan Borough of New York City, which is a highly urbanized region. Air was continuously sampled from a rooftop observatory that is situated ~90 m above sea level on one of the tallest buildings in the vicinity of the site. In June 2022, the Stonybrook PTR-ToF-MS was moved to the Flax Pond Marine Laboratory (40°57′36″N, 73°8′24″ W) near Stony Brook, New York, which is about 60 miles east of ASRC and located on the north side of Long Island in a forested suburban area. The Flax Pond Marine Laboratory is a 0.6 km$^2$ preserve that encompasses a tidal wetland area and is operated for research purposes by the School of Marine and Atmospheric Sciences of Stony Brook University. At Flax Pond, air was continuously sampled from a ~10 m tower.

### 2.2.Instrument Descriptions – PTR-ToF-MS

### 2.2.1. NOAA PTR-ToF-MS



The NOAA PTR-ToF-MS was deployed during SUNVEx, RECAP-CA, and FIREX-AQ. During FIREX-AQ, the NOAA PTR-ToF-MS used a traditional ion source and drift tube as described by Yuan et al. (2016). A full description of the operating parameters, VOC measurements, and calibration methods are provided by Gkatzelis et al. (2022).

During SUNVEx and RECAP-CA, the instrument was modified to use the Vocus focusing ion molecule reactor (TOFWERK, AG) and was operated following the recommendations by Krechmer et al. (2018). The Vocus provides greater sensitivity to VOCs compared to the traditional drift tube design due to the use of quadrupole ion guides that increase ion transmission. Here, the Vocus was operated at 2.5 mbar and with an axial electric field gradient of 65 V cm$^{-1}$ (E/N ~ 140 Td). The water flow to the ion source was maintained at 23 mL min$^{-1}$ and the drift tube was heated to 110°C. Typically, the quadrupole ion guide in the Vocus PTR-ToF-MS is operated at voltages > 275 V to reduce the transmission of reagent ions that would otherwise limit the lifetime of the detectors (Krechmer et al., 2018). Here, the quadrupole ion guide was tuned to 250 V to increase the transmission of ions produced from important VOCs with low molecular weights, such as ethanol (m/z 47), acetonitrile (m/z 42), and methanol (m/z 33). Figure S1 compares the product distribution of VOCs measured by the Vocus against those measured with the traditional drift tube. In general, the ion product distributions are comparable, though small differences in water clusters and fragmentation in the Vocus reflect the higher amount of water in the drift tube and a higher operating E/N. The degree of fragmentation in the NOAA Vocus PTR-ToF-MS is comparable to other PTR-MS systems with E/N > 120 Td (e.g., Buhr et al., 2002;Pagonis et al., 2019). Other Vocus PTR-ToF-MS instruments used in this study observed higher fragmentation owing to differences in operating conditions of the big-segmented quadrupole (BSQ). The implications of fragmentation from the BSQ are discussed further in Section 3.1.

To guide the identification of the proton-transfer-reaction products, a GC was used to trap and pre-separate ambient VOCs during SUNVEx. The GC deployed here is the same instrument used by Stockwell et al. (2021) to identify molecular isomers measured from coating headspaces. Briefly, the GC consists of a liquid nitrogen cryotrap coupled to a DB-624 column (Restek MXT-624; 30-m length × 0.25-mm inner diameter (I.D.), 1.4-µm film thickness). Samples were collected onto the cryotrap at predetermined volumes (typically 80 cm$^3$), then injected onto the column via rapid heating to 100°C. Nitrogen gas carried the sample through the column at 8 sccm while the column was heated from 40°C to 150°C at a rate of 12°C min$^{-1}$. The effluent from the column was injected into the PTR-ToF-MS inlet. In this study, we use this setup (termed GC-PTR-ToF-MS) to qualitatively assess isomer distributions and fragmentation patterns for VOCs detected during *in situ* sampling.

The GC-PTR-ToF-MS was primarily deployed during the ground-based sampling phase in Las Vegas, while PTR-ToF-MS only was used in Los Angeles. In GC-mode, samples were collected every 2 hours and automatically analyzed by PTR-ToF-MS. In between GC measurements, the PTR-ToF-MS sampled ambient air through a 10-m Teflon inlet at 2 L min$^{-1}$. During ambient sampling, instrument backgrounds were determined hourly by passing ambient air through platinum catalyst heated to 350°C. The PTR-ToF-MS was calibrated using gravimetrically-prepared gas standards, or by liquid calibration (Coggon et al., 2018).



When installed on the mobile laboratory, the PTR-ToF-MS sampled air through a 1-m Teflon inlet at 2 L min$^{-1}$, and instrument backgrounds were determined every 15 minutes. During an evening drive on 31 July 2021, the GC-PTR-ToF-MS was deployed to speciate VOCs on the Las Vegas Strip, where large crowds of people were present and anthropogenic emissions from personal care products, cooking, and other human activities were expected to be highest.

### 2.2.2. University of Oslo PTR-ToF-MS

The University of Oslo PTR-ToF-MS was deployed during FIREX-AQ to target $NH_3$, but also measured the same VOCs as the NOAA PTR-ToF-MS. The instrument was operated as described by Müller et al. (2014) with modifications to reduce the formation of $NH_4^+$ in the ion source as described by Tomsche et al. (2023). Briefly, the instrument sampled air through a heated inlet at a flowrate of 10–60 L min$^{-1}$ in order to reduce losses of $NH_3$ to inlet surfaces. The drift tube was operated at 2.1 mbar and 120°C with corresponding E/N ratio of 120 Td. VOC sensitivities were determined via calibrated using gravimetrically-prepared standards.

### 2.2.3. University of California Berkeley PTR-ToF-MS

The Berkeley Vocus PTR-ToF-MS (Aerodyne Research, Inc., Billerica, USA) was deployed during the RECAP-CA aircraft campaign on the US Navy Twin Otter. The PTR-ToF-MS was operated with a Vocus reactor set to 60°C, 2.0 mbar, and an E/N ratio of ~ 130 Td. The potential gradient along the drift tube was 590 V. The gradient between BSQ skimmer 1 and skimmer 2 was changed once during the campaign from 6 to 9.1 V, which resulted in an improved sensitivity for some VOCs, but significantly stronger fragmentation for other compounds such as nonanal (Fig. S2). Both operating conditions were calibrated. The reagent water flow was 20 mL min$^{-1}$. Similar to the NOAA-PTR-ToF-MS, the voltage of the quadrupole ion guide was operated at 200 V to improve the transmission for low-mass VOCs like methanol.

Ambient air was sampled via a 90 cm long heated (40°C) ¼-inch Teflon line through a Teflon filter from an isokinetic inlet (flow rate ~ 6 m s$^{-1}$ for 5 m length) with a mass flow controller at 1.5 L min$^{-1}$. Mass spectra were recorded at 10 Hz time resolution for a mass range of 10-500 Da. Zero-air blank measurements were conducted several times in each flight for 1-5 minutes during aircraft turns ~ 2-4 times per flight, each followed by a pulse of calibration gas ~ 1-5 minutes in duration. These in-flight calibrations were used to validate the sensitivities calculated from ground calibrations. Ground calibrations were conducted every 1-3 days (in total, 19 times) during the campaign using one of three gravimetrically prepared multicomponent VOC standards (Apel-Riemer Environmental Inc., Miami, FL, USA). More details on the instrument operation and calibration can be found in Pfannerstill et al. (2023).

### 2.2.4. Aerodyne PTR-ToF-MS

Aerodyne Research, Inc. deployed a Vocus PTR-ToF-MS during MOOSE 2021 (Krechmer et al., 2018;Riva et al., 2019). The Vocus was operated at a pressure of 2.2 mbar and axial voltage gradient of 600 V, corresponding to an E/N ratio of 125 Td. Data were recorded and processed at 1 Hz time resolution using the Tofware software (Aerodyne Research Inc. and TOFWERK) in Igor Pro (WaveMetrics) (Stark et al., 2015). Background measurements were conducted every 16



minutes by overflowing the Vocus inlet with air from a zero-air generator (ZAG) equipped with a Pt/Pd catalyst at 400°C. Calibrations were performed every 4 hours with a multicomponent VOC mixture (Apel-Riemer Environmental Inc., Miami, FL, USA; nominal 1 ppm in $N_2$) diluted with ZAG air. The sensitivities of species in the calibration mixture were correlated to their proton-capture-rate coefficient (Sekimoto et al., 2017). To calculate sensitivities for compounds not present in the calibration mixture, the slope of the linear fit was multiplied by the proton-capture-rate coefficient of the species of interest (Holzinger et al., 2019;Krechmer et al., 2018).

### 2.2.5. Stony Brook PTR-ToF-MS

The Stony Brook Ionicon high-resolution PTR-ToF-MS (Ionicon 8000, Analytik GmbH, Austria) was deployed in New York City during the COVID shutdown, and subsequently at Flax Pond on Long Island. In this study, the Stony Brook PTR-ToF-MS was operated with a drift field of ~130 Td, drift voltage of 600 V, reactor temperature of 60°C, and drift tube pressure of 2.3 mbar. The instrument sampled VOCs through a heated (60°C) 1/16-inch outer diameter (O.D.) capillary PEEK inlet (~1 m length) with bypass flow line teed off the 1/2-inch PTFE inlet line fitted with a blower on the back end (residence time of the gas was ~10 s). Data were collected at 1 Hz and integrated to 5-minute averages.

Calibrations were performed using a dynamic dilution system. VOC-free air was produced by pumping ambient air through a Pt-based catalytic converter at 400°C, then mixed with multicomponent gas calibration mixture (Apel-Riemer Environmental Inc., Miami, FL, USA) that included isoprene. Calibration was performed spanning a concentration range of observed values (0, 5, 10, 15, 20 ppbv). At ASRC, the calibration gas was typically analyzed twice a week prior to the COVID-19 lockdown, and typically every 1-2 weeks during the lockdown given limited access to the observatory. At Flax Pond, the calibration gas was analyzed once a week.

### 2.3. Instrument Descriptions – Gas Chromatography

### 2.3.1. NOAA GC-MS

The NOAA GC-MS provided speciated VOC measurements during SUNVEx, RECAP-CA, and FIREX-AQ. During RECAP-CA, the NOAA GC-MS was deployed to the ground site to sample ambient air on a 20 minute duty cycle. The GC-MS collects two separate 240 mL samples and analyzes each on two channels. Channel 1 consists of a $CO_2$ trap (Ascarite II, Thomas Scientific), a water trap operated at -55°C, and a sample trap operated at -165°C. The series of traps are linked to an $Al_2O_3$-KCl porous layer open tubular column (Restek RT-Alumina bonded porous polymer/KCl; 30-m length × 0.25-mm I.D., and 4-µm film thickness) designed to separate light hydrocarbons. Channel 2 consists of the water and sample traps, but is coupled to a DB-624 column identical to the column used in the GC-PTR-ToF-MS. This column separates hydrocarbons up to $C_{12}$, as well as select oxygen-, halogen-, and nitrogen-containing VOCs. The effluent of each column is analyzed using a quadrupole mass spectrometer (Agilent 5975C) operated in selected ion monitoring/scan mode. The GC-MS was calibrated using a gravimetrically-prepared gas mixture containing 50 VOC components.

VOCs analyzed by NOAA GC-MS in Las Vegas and during FIREX-AQ were first sampled using a whole air sampling canister system (iWAS, Lerner et al., 2017). The iWAS system consists of a





stainless steel compressor and 24 2.7 L electropolished stainless steel canisters. During SUNVEx, canisters were filled every 2 hours during stationary ground-based sampling, while targeted samples were taken during mobile drives. During FIREX-AQ, targeted samples were filled on demand up to a total number of 72 per flight. The canisters were shipped to either Boulder, CO, or Pasadena, CA, and analyzed by GC-MS within four days of collection to minimize sampling artifacts.

### 2.3.2. UCI WAS

The University of California, Irvine Whole Air Sampler (UCI WAS) was deployed on the DC-8 during FIREX-AQ to sample VOCs and halocarbons. The UCI WAS operated similarly to the NOAA canister system, where samples were collected into electropolished stainless steel canisters, then analyzed offline within seven days using a series of laboratory GC systems. The operation of the airborne WAS and laboratory GCs is fully described elsewhere (Colman et al., 2001;Simpson et al., 2020;Simpson et al., 2010). All samples were analyzed on a multi-column GC system coupled to flame ionization, electron capture, and mass selective detectors. The whole system is calibrated using a suite of VOC standards.

### 2.3.3. TOGA-TOF

The NCAR Total Organic Gas Analyzer with a TOFWERK electron ionization high-resolution time-of-flight mass spectrometer (TOGA-TOF) was deployed on the DC-8 during FIREX-AQ to provide *in situ* GC measurements of a large suite of VOCs including hydrocarbons, oxygenated VOCs (OVOCs), and halogen-, nitrogen- and sulfur-containing VOCs. A full description of the TOGA system is provided by Apel et al. (2015). Briefly, during FIREX-AQ the TOGA-TOF continuously sampled 13-mL aliquots of ambient air for approximately 35 seconds every 105 seconds, concentrating the VOCs in two cryogenic preconcentration steps prior to injection onto a Restek MTX-624 column (I.D. = 0.18 μm, length = 8 m). Helium gas carried the samples through the column, which was heated from 25–120°C at a rate of 100°C min$^{-1}$, and the effluent was analyzed by the electron ionization time-of-flight mass spectrometer. The TOF-MS was operated at 70 eV and nominal mass resolution 3000 Δm m$^{-1}$. The system was calibrated several times per flight and in the laboratory before and after the campaign using a series of multicomponent VOC standards (Apel-Riemer Environmental Inc., Miami, FL, USA).

### 2.3.4. Aerodyne GC-MS

The ARI GC-MS system consists of three main components: (1) a thermal desorption preconcentrator (TDPC) (Aerodyne Research, Inc.) for sample collection, (2) a GC (Aerodyne Research, Inc.) for sample separation, and (3) an electron ionization time-of-flight mass spectrometer (EI-ToF-MS) (TOFWERK AG, model EI-HTOF) for sample detection (Gilman et al., 2013; Obersteiner et al., 2016; Claflin et al., 2021). For the MOOSE campaign, the TOF-MS was operated at 70 eV and nominal mass resolution 3000 Δm m$^{-1}$. The GC is a 2-channel system, where both separation channels use identical preconcentration steps. The TDPC employed for this campaign relied upon two-stage adsorbent trapping for preconcentration of analytes. The sample



is initially collected onto multibed (Tenax TA/Graphitized Carbon/Carboxen 1000), preconditioned glass sorbent tubes that are optimized for $C_{2/3}$–$C_{30}$ species. The first stage of trapping allows sampling rates up to 100 sccm, followed by forward-flushing with UHP nitrogen to remove water. For the higher volatility channel (Channel 1), additional $H_2O$ was removed from the ambient sample stream via passing through a cooled PFA tube (1/8-inch O.D., 1/16-inch I.D., 3.375-inch length, 15°C) prior to trapping of VOCs to avoid water condensed in the sample tube. After the initial collection and water purge, the sample was then transferred to the focusing stage, which is a multibed (Tenax, Carbopack X, Carboxen 1003) glass cold trap. After preconcentration, the samples are transferred to different separation columns (Restek Rt-Q-Bond and Rxi-624) for Channels 1 and 2, respectively. Channel 1 is optimized for separation of $C_3$-$C_4$ alkanes, $C_1$-$C_2$ halocarbons and $C_1$-$C_3$ oxygenates; Channel 2 is optimized for $C_5$-$C_{12}$ alkanes, $C_6$-$C_{10}$ aromatics, $C_3$-$C_6$ oxygenates.

The GC inlet consisted of approximately 8 m of 0.25-inch O.D., 0.15625-inch I.D. PFA tubing connected to a sample pump, with an inlet flow of approximately 1 slpm. The GC pulled a sub-flow from the main GC inlet via 1-m length, 0.125-inch O.D., 0.0625-inch I.D. PFA tubing. The sample flow was 80 sccm to each GC channel for 10 minutes during each 30-minute analytical cycle. Before preconcentration, the ambient sample is passed through a bed of pre-cleaned sodium sulfite (nominal 1 g) to scrub ozone and thereby reduce sampling artifacts [Helmig, 1997].

The TDPC employed for this campaign relied upon two-stage adsorbent trapping for preconcentration of analytes. The sample is initially collected onto multibed (Tenax TA/Graphitized Carbon/Carboxen 1000), preconditioned glass sorbent tubes that are optimized for $C_{2/3}$–$C_{30}$ species. The first stage of trapping allows sampling rates up to 100 sccm, followed by forward-flushing with UHP nitrogen to remove water. For the higher volatility channel (Channel 1), additional $H_2O$ was removed from the ambient sample stream via passing through a cooled PFA tube (1/8-inch O.D., 1/16-inch I.D., 3.375-inch length, 15°C) prior to trapping of VOCs to avoid water condensed in the sample tube. After the initial collection and water purge, the sample was then transferred to the focusing stage, which is a multibed (Tenax, Carbopack X, Carboxen 1003) glass cold trap.

Calibration was performed by multicomponent VOC calibrant gas (Apel-Riemer Environmental Inc., Miami, FL, USA) diluted in UHP N2 at sufficient total flow to overflow the GC sub-inlet. Instrument calibrations were performed in-field each night throughout the campaign via automated valve switching. Note that this calibration mixture was the same as described in Section 2.2.3.

## 3. Results

The following sections outline PTR-ToF-MS interferences observed for ions typically assigned to isoprene, oxygenated VOCs, and aromatic VOCs. The primary data used for this analysis are from the NOAA PTR-ToF-MS, which provided direct evidence of interferences via GC pre-separation. Each section begins with a description of GC-PTR-ToF-MS samples collected along the Las Vegas Strip during SUNVEx. This region comprises many hotels and entertainment establishments and is impacted by emissions from fossil fuels, VCPs, and restaurant cooking. This region had the highest observed mixing ratios compared to New York, Detroit, and Los Angeles. These interferences are then compared to ground site data collected by the NOAA PTR-ToF-MS during



SUNVEx/RECAP-CA. Finally, other observations are used to show the ubiquity of these
interferences across instruments and urban environments. A key focus of this discussion is the
impact of fragmentation on PTR-ToF-MS observations of m/z 69 (exact mass 69.070), which is
typically assigned to isoprene. We also focus on methods to determine urban isoprene mixing
ratios when fragmentation from other higher molecular weight molecules is important.
### 3.1.    Isoprene
### 3.1.1.  Known interferences to isoprene (m/z 69)
Biogenic VOCs are commonly reported by PTR-ToF-MS, including isoprene and the sum of
monoterpene isomers. Isoprene is the dominant biogenic VOC emitted by urban foliage and is a
major contributor to urban OH reactivity (Calfapietra et al., 2013). Interferences to isoprene in
PTR-ToF-MS spectra results from the production of the $C_5H_8H^+$ ion, which is a common fragment
for higher-carbon aldehydes (> C$_5$), alkenes, and cycloalkanes (Buhr et al., 2002;Gueneron et al.,
2015;Pagonis et al., 2019;Romano and Hanna, 2018). Previous studies have characterized ambient
isoprene interferences from 2-methyl-3-buten-2-olalkenes emitted from biogenic sources (e.g.,
Karl et al., 2012) and cycloalkanes emitted from fossil fuels use and oil and natural gas production
(e.g., Gueneron et al., 2015;Warneke et al., 2014;Pfannerstill et al., 2023). For example, Gueneron
et al. (2015) showed that substituted cyclohexanes and cyclohexenes produce fragmentation
patterns that consist largely of m/z 111, m/z 125, m/z 69, m/z 83, m/z 57, and other lesser-abundant
hydrocarbon fragments. In regions with significant oil and natural gas development, these
compounds may produce interferences at m/z 69 which can interfere with the signal resulting from
biogenic sources of isoprene (e.g., Warneke et al., 2014). Similarly, Kilgour et al. (2021) show
that aldehydes emitted due to ozone deposition to surface ocean waters may interfere with the
quantification of isoprene. The key aldehydes observed to produce an interference were nonanal
and octanal. The same aldehydes may be produced on inlet surfaces exposed to high ozone
concentrations and result in an isoprene artifact (Ernle et al., 2023;Vermeuel et al., 2022).
### 3.1.2.  Characterizing aldehyde interferences to m/z 69 using GC-PTR-ToF-MS
Figure 2 shows a GC-PTR-ToF-MS chromatogram of the ion typically assigned to isoprene (m/z
69, $C_5H_8H^+$). This sample was collected on the Las Vegas Strip in the evening (~ 22:15 local time,
LT) when biogenic emissions of isoprene are expected to be low. The chromatogram shows that
isoprene (retention time, RT = 65 s) is only a small contributor to the signal at m/z 69 measured in
this region. Additional peaks are observed at RT = 210 s, 250 s, 600 s, and 710s. These peaks are
not cycloalkanes, as might be expected from mobile source emissions (Gueneron et al., 2015);
rather, these are dehydration and fragmentation products of saturated aldehydes, including
methylbutanal, pentanal, octanal, and nonanal. Chromatograms of the parent ions attributed to
octanal and octanone (m/z 129, $C_8H_{16}OH^+$), together with nonanal and nonanone (m/z 143,
$C_9H_{18}OH^+$) are shown in Fig. S3. The parent ion and the dehydration products (m/z 111 $C_8H_{15}^+$
and m/z 125 $C_9H_{17}^+$, respectively) are observed, but at different ratios between the aldehydes and
ketones. Pentanal and methylbutanal almost entirely dehydrate and do not exhibit significant signal
at the parent ion mass. Comparisons of ambient observations with GC-PTR-ToF-MS
chromatograms of standard mixtures show that only aldehydes (and not the ketones) are observed
in significant quantities on the Las Vegas Strip (Fig. S3).




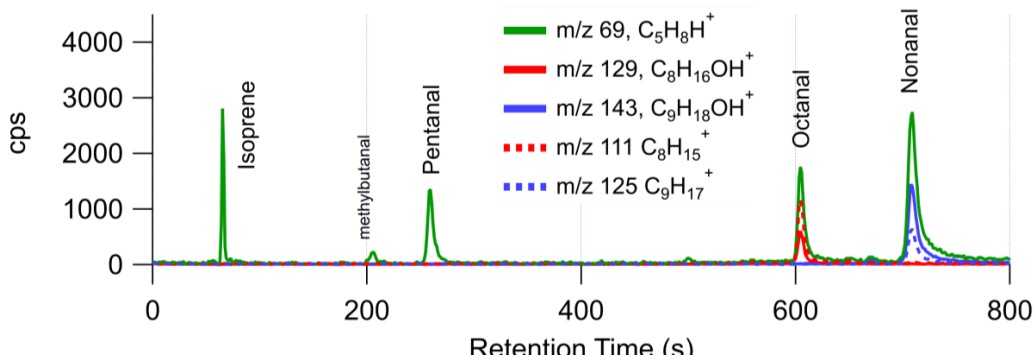

**Figure 2.** GC-PTR-ToF-MS chromatogram from downtown Las Vegas at 22:15 on 30 July 2021, showing the
contributions of isomers and fragments to the ion typically assigned to isoprene (m/z 69, $C_5H_8H^+$).

These aldehydes emissions likely result from cooking (Arata et al., 2021;Klein et al., 2016;Schauer
et al., 1999;Karl et al., 2018;Wernis et al., 2022) and their significant presence on the Las Vegas
Strip possibly reflects the high density of restaurants along Las Vegas Boulevard. Figure 3 shows
mobile laboratory measurements of nonanal and m/z 69 during evening drives on 28 June and 30
July 2021. GC-PTR-ToF-MS sampling was only conducted on 30 July and the location of the
sample (Las Vegas Strip) is shown in Fig. 1a. During both drives, m/z 69 was enhanced along Las
Vegas Boulevard and mixing ratios reached a maximum of 6 ppb (assuming a sensitivity
equivalent to isoprene). On 28 June, m/z 69 and nonanal detected at m/z 143 ($C_9H_{18}OH^+$) are
highly correlated ($r^2 > 0.93$), suggesting that these ions share a common source. A similar
correlation was observed between m/z 69 and octanal ($r^2 = 0.90$).

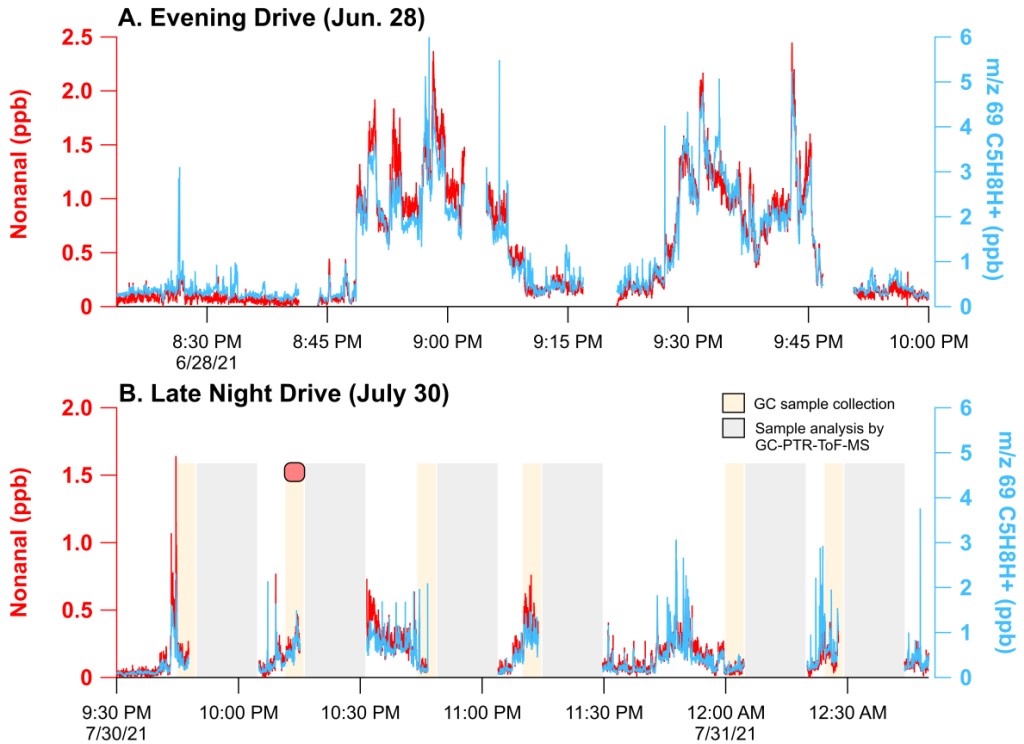

**Figure 3.** Mobile laboratory data showing PTR-ToF-MS measurements of nonanal (m/z 143, $C_9H_{18}OH^+$, red) and m/z 69 ($C_5H_8H^+$, blue) on the Las Vegas Strip during nighttime hours on (a) 28 June and (b) 30 July 2021. GC samples were only collected on 30 July, and the shaded regions in (b) show periods of sample collection (beige) and sample analysis (grey). The red marker in (b) indicates the time of the GC-PTR-ToF-MS sample shown in Fig. 2.

Long-chain aldehydes are not routinely reported in urban datasets and the isoprene interference due to aldehyde fragmentation is underappreciated in ambient PTR-ToF-MS datasets. Studies have described how aldehydes produced on the surfaces of inlet tubing interfere with isoprene measured by PTR-ToF-MS in remote forests (Vermeuel et al., 2022) and in the stratosphere (Ernle et al., 2023), and aldehydes emitted from ocean surface waters also interfere with isoprene measurements in laboratory studies and ambient measurements near coastal regions (Kilgour et al., 2021). Long-chain aldehydes are likely ubiquitous in cities, and cooking activities are likely a major source of octanal and nonanal resulting in an isoprene interference (Wernis et al., 2022;Peng et al., 2022).

The interference from aldehydes is likely common across PTR designs, even though differences could exist due to operating conditions (e.g., the E/N ratio). Figure S1 compares the fragmentation patterns of pentanal, octanal, and nonanal observed in the NOAA PTR-ToF-MS (E/N ~ 140 Td), which utilizes the Vocus ion source, to those reported by Buhr et al. (2002) (E/N ~120–130 Td), which employ a traditional drift tube and quadrupole. In both reactor designs, $C_5$-aldehydes dehydrate to produce $C_5H_8H^+$ (m/z 69) directly, while larger aldehydes such as octanal and nonanal dehydrate to produce $C_8H_{15}^+$ (m/z 111, exact mass: 111.117) and $C_9H_{17}^+$ (m/z 125, exact mass: 125.123), then further fragment to produce the $C_5H_8H^+$ ion (Buhr et al., 2002;Pagonis et al., 2019).



These dehydration products are unique to aldehydes since ketone isomers do not undergo
significant fragmentation (Fig. S3).
We note that even though the fragmentation was similar between two instruments with different
reactor designs, fragmentation may result from the operation of other instrument components. For
example, the intensity of aldehyde fragmentation was found to vary strongly with voltage gradients
within the BSQ of the Berkeley Vocus-PTR-ToF-MS (Fig. S2). These results show that PTR-ToF-
MS systems employing quadrupole ion guides may exhibit fragmentation outside of the drift tube
region; consequently, care may be needed when tuning instruments to minimize unwanted
secondary reactions.

### 3.1.3. Corrections to ground site observations of m/z 69 in Los Angeles and Las Vegas

GC-PTR-ToF-MS and mobile laboratory measurements described in Section 3.1.2 indicate that
aldehydes significantly contribute to the signal at m/z 69 in urban areas. Chromatograms show that
the dehydration products from nonanal (m/z 125) and octanal (m/z 111) are useful markers that
distinguish aldehydes from ketone isomers. Coincidentally, the dehydration products from nonanal
and octanal are identical to the fragments produced from substituted cyclohexanes, which interfere
with isoprene in hydrocarbon-rich environments (see Gueneron et al., 2015;Warneke et al.,
2014;Pfannerstill et al., 2023). Here, it is proposed that the signals at m/z 111 and m/z 125 can be
used as proxies to calculate the contribution from aldehyde and cycloalkane fragmentation on the
signal at m/z 69 in urban areas.
Figure 4 shows how the sum of m/z 111 and m/z 125 (termed the "isoprene interference") varies
relative to the signal at m/z 69 measured at the ground sites in Los Angeles and Las Vegas. In Los
Angeles, high daytime emissions of isoprene dominate and comprise most of the signal of m/z 69
from 6:00–19:00 LT. The high variability in the signal at m/z 69 is caused by very localized
emissions from trees upwind of the measurement site. In Las Vegas, isoprene emissions are much
lower and the diel pattern of m/z 69 closely follows the behavior of the isoprene interference with
only small additional signal during the daytime.



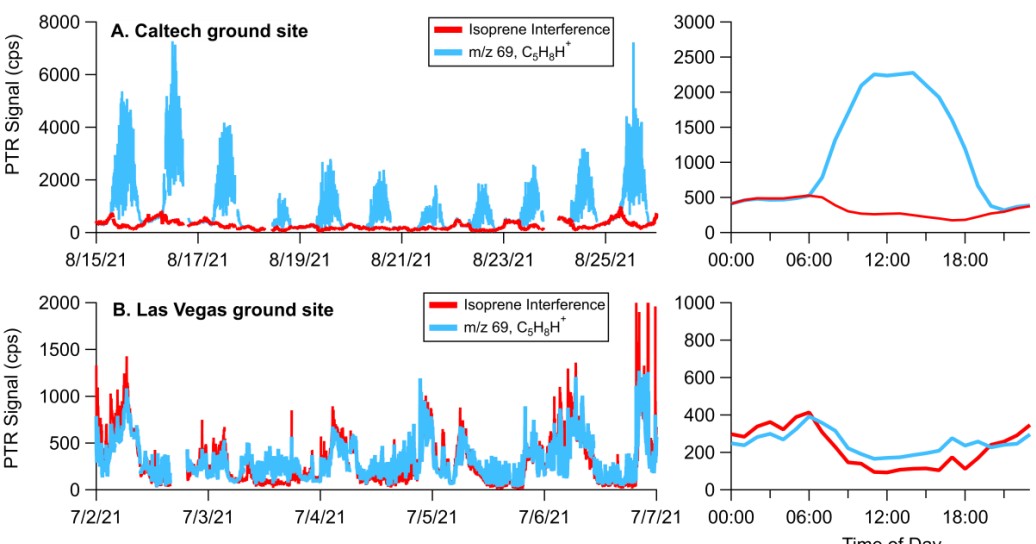

**Figure 4.** Time series and diurnal pattern of the signal at m/z 69 ($C_5H_8H^+$) and the isoprene interference (m/z 111 +
m/z 125) measured at (a) the Caltech ground site and (b) the Las Vegas ground site. The time series data are shown
for select periods to illustrate correlations between the isoprene interference and m/z 69. The diel patterns on the right
are campaign averages.
Biogenic isoprene is predominantly emitted during daytime hours (e.g., Guenther et al., 2012),
while the isoprene interference in both cities is more prevalent at night. These differences in diurnal
patterns can be leveraged to subtract interferences from aldehydes and cycloalkanes from PTR-
ToF-MS measurements of m/z 69. Here, the signals at m/z 69, m/z 111, and m/z 125 are analyzed
between 00:00-04:00 LT when daytime isoprene from biogenic sources is expected to be low. The
instrument response to aldehyde and cycloalkane fragmentation is calculated by determining the
ratio of m/z 69 to the sum of m/z 111 + m/z 125. This ratio is then applied to the full dataset
following Eq. 1.

$$\text{m/z } 69_{\text{Corrected}} \ = \ S_{69} - S_{111+125} \cdot f_{69/(111+125)} \ \text{(Eq. 1)}$$

$S_{69}$ is the signal measured at m/z 69, $S_{111+125}$ is the signal of the isoprene interference (sum of m/z
111 + m/z 125), and $f_{69/(111+125)}$ is the interference ratio determined at night. The nighttime
interference ratio is 3.0 in Las Vegas and 3.5 in Los Angeles (Fig. S4). The differences between
the cities may reflect variations in the distribution of aldehydes and cycloalkanes.
Figure 5 shows how measurements of m/z 69 change as a result of this correction and compares
the corrected/calculated isoprene mixing ratios to the GC-MS measurements co-located with the
PTR-ToF-MS. In Los Angeles, the correction largely impacts m/z 69 signals at night. The diurnal
pattern shows that average mixing ratios approach zero in the evenings, though increases in
nighttime isoprene mixing ratios are observed during some periods (e.g., 22–24 August, Fig. S5).
Corrections to m/z 69 during the daytime lead to a ~ 10% decrease in reported mixing ratios. This
shows that even when isoprene emissions are high, VOC fragmentation can have a significant
impact on the signal at m/z 69.




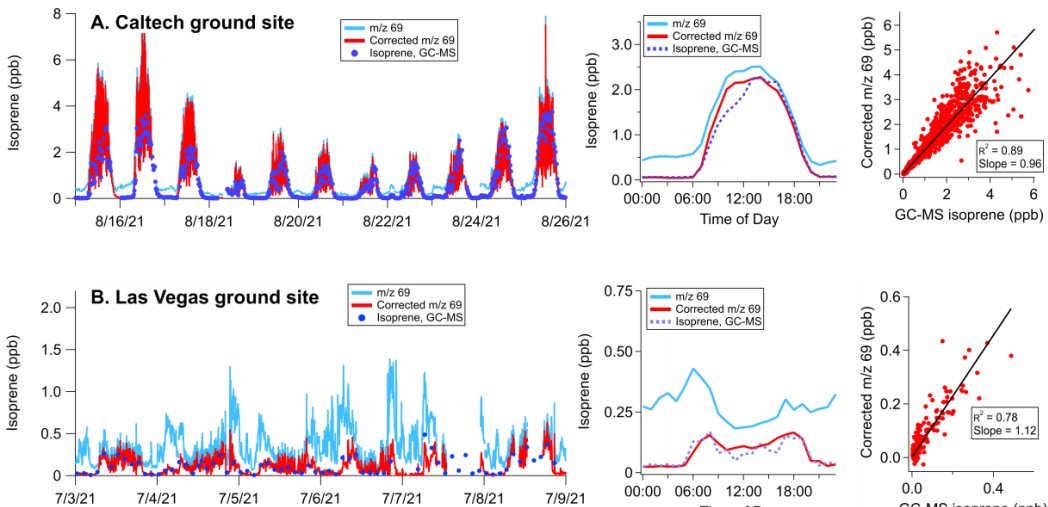

**Figure 5**. Uncorrected and corrected m/z 69 as time series, diel averages, and correlation plots for (a) the Caltech and
(b) Las Vegas ground sites. GC-MS measurements are shown for comparison against the corrected m/z 69 isoprene
mixing ratios. A detailed comparison of nighttime isoprene corrections in Los Angeles is shown in Fig. S5.
The corrected m/z 69 measurements are well correlated with GC-MS isoprene measurements ($r^2$
$= 0.89$) and agree to within 4%. At high isoprene mixing ratios, the measurements exhibit a greater
degree of scatter. This variability likely results from the differences in sampling timescales (1 s for
PTR-ToF-MS, ~ 120 s for GC-MS) along with the high variability of isoprene emissions from
trees at the measurement site. When averaged to a diel profile, the daytime mixing ratios also agree
to within 4%. Both instruments show that average isoprene decreases to low mixing ratios at night
($< 0.05$ ppb). The GC-MS observed a number of periods of enhanced nighttime isoprene, likely
from non-biogenic sources. Remarkably, after accounting for the isoprene interference, the
corrected m/z 69 mixing ratios from the PTR-ToF-MS captures the variability in nighttime
isoprene observed by GC-MS in Los Angeles (Fig. S5). On average, the isoprene interference
represents ~90% of the nighttime signal of m/z 69.
The isoprene correction is most impactful on the Las Vegas measurements where isoprene
emissions are low and aldehydes + cycloalkane fragments constitute a larger fraction of the signal
at m/z 69. Without correction, the variability in m/z 69 across all daytime hours is driven by the
isoprene interference (Fig. 4). After the interference contributions are subtracted, corrected
isoprene mixing ratios approach zero at night and decrease by nearly 50% to 0.1–0.15 ppb during
the day (Fig. 5b). The resulting diel pattern changes substantially and exhibits a daytime peak that
is consistent with the expected pattern for isoprene. GC-MS measurements show that isoprene
mixing ratios were typically $< 0.2$ ppb and the corrected m/z 69 diel pattern generally matches the
average diel pattern of isoprene reported by GC-MS. Though the number of canister samples in
Las Vegas were limited (total 275, sampled every 2–4 h), a comparison between the corrected m/z
69 and GC-MS isoprene shows that the measurements agree to within 15%.
**3.1.4.  Corrections to aircraft measurements of  m/z 69 over Los Angeles**



The isoprene interferences observed during SUNVEx and RECAP-CA show that PTR-ToF-MS measurements of m/z 69 are significantly impacted by aldehydes and cycloalkanes. To assess the impact of isoprene interferences at higher altitudes, we analyze the FIREX-AQ and RECAP-CA measurements of flights in the Los Angeles Basin and determine corrected m/z 69 signals following Eq. (1). One challenge to this approach is that the DC-8 and Twin Otter aircraft did not sample the Los Angeles Basin at night, and therefore the interference ratio ($f_{69/(111+125)}$) is not easily determined in the absence of isoprene. To overcome this limitation for FIREX-AQ, we vary the interference ratio until the corrected m/z 69 signals reported by the NOAA PTR-ToF-MS matches with the isoprene mixing ratios reported by GC instrumentation on the DC-8. The resulting ratio determined by iteration (4.4) is ~ 20% higher than the ratio determined at ground level during SUNVEx (3.5), which likely reflects differences between the operating conditions and drift tube designs used on the NOAA PTR-ToF-MS during FIREX-AQ and SUNVEx (traditional vs. Vocus).

Figure 6 illustrates the spatial and temporal variability in (a) the corrected m/z 69 mixing ratios and (b) the calculated interference. Transits to the north show that the interference is highest along the San Gabriel Mountains where anthropogenic pollution typically builds in the Los Angeles Basin (Angevine et al., 2013) and reached mixing ratios as high as 500 ppt. The interference correlates well with both methylcyclohexane measured by the UCI WAS and methylpropanal measured by TOGA-TOF, which are proxies for the species known to fragment to produce the isoprene interference (i.e., cycloalkanes and aldehydes). Corrected m/z 69 mixing ratios only exhibit significant enhancements in regions where the DC-8 sampled air close to vegetation. Short bursts of isoprene were observed above the San Gabriel Mountains, but mixing ratios were typically lower than 500 ppt. Over the entire flight, the isoprene interference constituted > 50% of the signal observed at m/z 69.

Figure 6c compares the PTR-ToF-MS measurements to GC-based samples for uncorrected m/z 69 (top) and corrected m/z 69 mixing ratios (bottom). These comparisons show that the isoprene interference resulted in an overestimation of PTR-ToF-MS measurements of isoprene by at least a factor of 2. At times, the NOAA PTR-ToF-MS measured mixing ratios of m/z 69 that were 5 times larger than the isoprene mixing ratios reported by GC-based methods. The Oslo PTR-ToF-MS also sampled onboard the DC8 during FIREX-AQ and presented an opportunity to compare to the fragmentation observed by the NOAA PTR-TOF-MS. Following a similar correction procedure as described above, Fig. S6 shows that the Oslo PTR-ToF-MS measured the same degree of interferences as the NOAA PTR-ToF-MS (i.e., fragmentation biased isoprene measurements high by at least a factor of 2). The consistency between both instruments demonstrates that isoprene interferences are common across PTR-ToF-MS designs (i.e., Tofwerk vs. Ionicon).

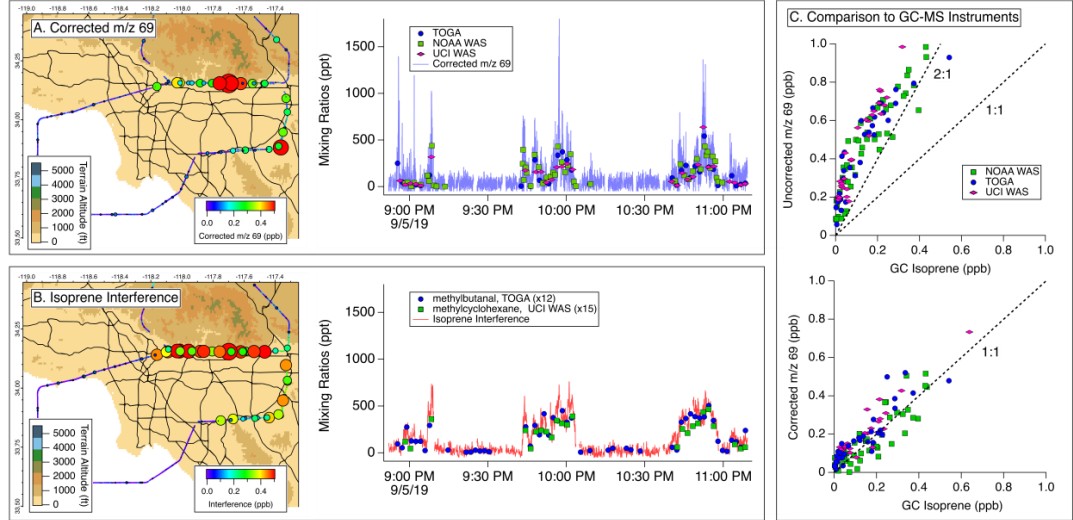

**Figure 6.** Impact of isoprene interference correction on m/z 69 measurements from the NOAA PTR-ToF-MS during FIREX-AQ. (a) Map of corrected m/z 69 distribution (left) and time series with corresponding measurements of isoprene from GC-MS samples (right). (b) Map of isoprene interference (left) and time series with GC-MS measurements of methylcyclohexane and methylpropanal, which are proxies for cycloalkanes and aldehydes known to contribute to the signal at m/z 69. (c) Comparisons of PTR-ToF-MS measurements of m/z 69 and GC-based isoprene mixing ratios for uncorrected m/z 69 (top) and corrected m/z 69 (bottom) using Eq. (1) with an interference ratio = 10.

The Berkeley Vocus PTR-ToF-MS also observed interferences to m/z 69 during the RECAP-CA flights. Unlike FIREX-AQ, GC-MS measurements were not available onboard the Twin Otter to compare against PTR-ToF-MS measurements. To evaluate the interference contributions to m/z 69 here, we determine the interference ratio from nonanal calibrations and compare it with data collected from the Central Valley and Los Angeles Basin. In these regions, the signal at m/z 69 and the sum of m/z 111 + 125 are well-correlated with a slope that closely matches the measured fragmentation pattern for nonanal (Fig. S7). The interference ratio observed in the Central Valley where oil and natural gas emissions are significant is similar to the ratio observed in Los Angeles where aldehydes are more important. In the Central Valley, periods of high biogenic influence are clearly separated from periods of high interference from anthropogenic emissions. Building on these responses, we use the calibrated data for nonanal to derive an m/z 69 correction in Los Angeles. This method is limited in accounting for molecules that have fragmentation ratios differing from nonanal, but since the interference ratio observed from oil and gas regions and from the Los Angeles Basin is similar to the nonanal fragmentation ratio, we expect the uncertainty to be relatively small despite there being no GC comparison.

Figure 7 shows the impact of the isoprene interference on the Berkeley PTR-ToF-MS data. The Twin Otter flew nine flights and the total signal of m/z 69 varied between 200–1200 ppt. Similar to the observations by the NOAA PTR-ToF-MS during FIREX-AQ, the isoprene interference during RECAP-CA was at least 50% of the signal observed at m/z 69 (Fig. 7c). The Twin Otter sampled a larger swath of area than the DC-8, and Fig. 7 shows that the interference is persistent across the Los Angeles Basin (Fig. 7b) at mixing ratios as high as 600 ppt. Similar to the DC-8 flight, corrected m/z 69 mixing ratios are highest along the San Gabriel Mountains. The Twin Otter





is capable of sampling at lower altitudes than the DC-8, and therefore larger mixing ratios of
isoprene were observed.

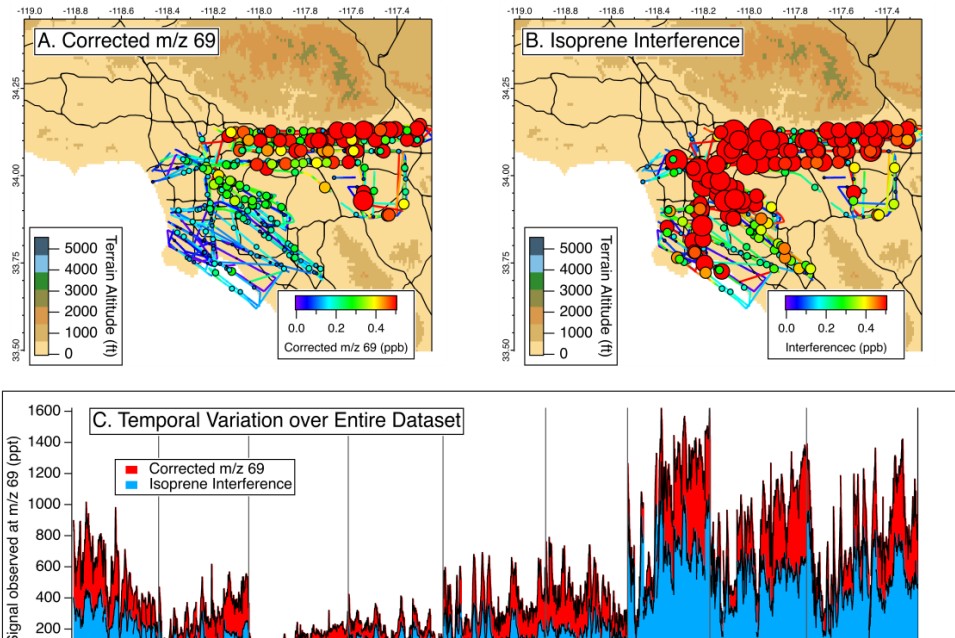

**Figure 7.** Impact of isoprene interference correction on isoprene measurements during RECAP-CA as (a) the corrected
isoprene distribution, (b) the isoprene interference, and (c) a pseudo time series of the total m/z 69 signal colored by
the contributions of corrected m/z 69 and isoprene interference for all flights.

The contribution of the isoprene interference observed from the aircraft with the Berkeley PTR-
ToF-MS differs from the observations at Caltech during RECAP-CA. On the ground, the isoprene
interference was ~10% of the signal at m/z 69 during daytime hours (Fig. 5a), while at altitude it
was > 50%. This difference can be explained by (1) the abundance of isoprene emitters close to
the ground site, (2) the differences in reactivities between isoprene, aldehydes, and cycloalkanes,
and (3) the different instrument setting of the Berkeley PTR-ToF-MS (Fig. S2). Isoprene is highly
reactive towards atmospheric oxidants such as the OH radical ($k_{OH}$ ~ $1 \times 10^{-10}$ cm$^3$ molecule$^{-1}$ s$^{-1}$
), whereas saturated aldehydes and cycloalkanes are expected to be 5–10 times less reactive
(Burkholder et al., 2019). This difference in reactivity may alter the distribution of VOCs that
contribute to m/z 69 and result in higher interferences aloft. The DC-8 and Twin Otter aircraft did
not specifically target altitude profiling while sampling in the Los Angeles Basin, but future work
may help to characterize the impact of the isoprene interference at other altitudes.

**3.1.5.  Corrections to m/z 69 measured in Detroit, MI**

The SUNVEx/RECAP-CA/FIREX-AQ data reflect the behavior of the NOAA, Oslo, and Berkeley
instruments during summertime measurements in Los Angeles. Isoprene interferences likely





impact ground and airborne measurements in other cities and at other times of year. Figure 8 shows
the impact of interferences to the signal at m/z 69 reported by the Aerodyne PTR-ToF-MS
measurements during MOOSE. This campaign targeted emissions in Detroit, MI, where the
Aerodyne Mobile Laboratory conducted a mix of mobile and stationary sampling in select
locations around the metropolitan area (Fig. 1). Figure 8a shows a subset of PTR-ToF-MS
measurements of m/z 69 and corrected m/z 69, along with isoprene measurements by the Aerodyne
GC-MS. Corrected m/z 69 is calculated by determining the interference ratio at night (= 2.54),
similar to the approach used to calculate interferences during SUNVEx/RECAP-CA. Figure 8b
shows normalized histograms for each measurement over the entire deployment.

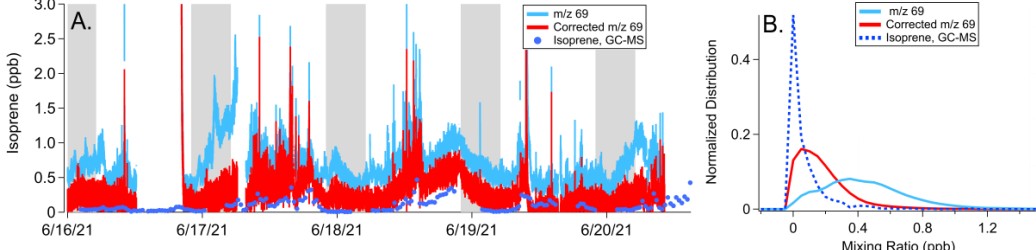

**Figure 8.** Impact of isoprene interference correction on the PTR-ToF-MS data collected from the Aerodyne Mobile
Laboratory during the MOOSE campaign as a (a) time series of GC-MS measurements, uncorrected and corrected
m/69 isoprene mixing ratios. The shaded regions show nighttime measurements (22:00–05:00 LT). (b) Histograms
showing the distribution of m/z 69 measured by the Aerodyne PTR-ToF-MS, corrected m/z 69 isoprene mixing ratios,
and isoprene mixing ratios measured by the GC-MS.

Without correction, m/z 69 reported by PTR-ToF-MS exhibits a broad distribution with a peak
mixing ratio of ~ 0.4 ppb (Fig. 8b). After applying the corrections described by Eq. (1), m/z 69
signals decrease by nearly a factor of 2 and show better agreement with isoprene reported by GC-
MS. Corrected m/z 69 mixing ratios are still a factor of 2 higher than the isoprene mixing ratios
reported by the GC-MS. One possibility is that other VOCs in the Detroit region may also
contribute to the signal at m/z 69. GC pre-separation measurements directly using PTR-ToF-MS
were not conducted during MOOSE; consequently, it is difficult to determine what other species
might contribute to m/z 69 in this region. Broad deployment of GC-PTR-ToF-MS measurements
in urban areas may help to better quantify the contributions of fragmenting species to PTR-ToF-
MS measurements of m/z 69.

**3.1.6. Seasonal changes to the m/z 69 interferences observed in New York City**

Figure 9 shows the impact of the isoprene interference on data reported by the Stony Brook PTR-
ToF-MS during a year-long sampling effort to characterize emissions in New York City during
the COVID-19 lockdown. Shown here are mixing ratios of m/z 69 along with the estimated
contribution to m/z 69 from the isoprene interference. We calculate the isoprene interference for
each season, and present the diurnal patterns in the top row. The interference ratio ($f_{69/(111+125)}$, Eq.
(1)) is similar in spring, summer, and winter (2.2–2.5), but lower during fall (1.9).



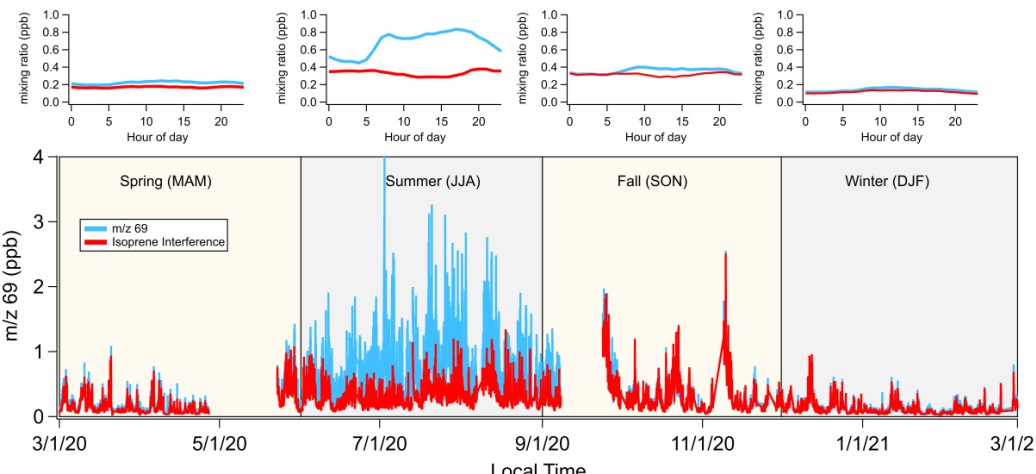

**Figure 9.** (bottom) Time series of m/z 69 and isoprene interference measured by the Stony Brook PTR-ToF-MS at
the urban ASRC ground site in New York City. (top) Diel patterns of m/z 69 and isoprene interference mixing ratios
for each season.
The signal at m/z 69 is variable across seasons and the highest mixing ratios are observed during
summer. The isoprene interference is a major contributor to m/z 69 in fall, winter, and spring (77-
88% of total signal) and strongly influences the day-to-day variability. During summertime
isoprene emissions from urban foliage increases the variability in m/z 69 and results in higher
mixing ratios of m/z 69 during the day. The isoprene interference increases the background mixing
ratios of m/z 69 and dominates the total signal at night.
The ASRC site is located in a heavily urbanized region and the PTR-ToF-MS sampled air at the
top of the building where mixing ratios of isoprene are likely lower. The persistent, high
contribution from the isoprene interference to m/z 69 during all seasons likely reflects the high
emissions of aldehydes and cycloalkanes from anthropogenic sources in this region. Figure S8
contrasts the measurements at ASRC with those reported from the Flax Pond site. Flax Pond is
located in a less-densely populated region of Long Island where biogenic sources of isoprene are
more abundant. There, interferences are a much smaller fraction of the signal at m/z 69 ( < 10%)
and the variability is largely driven by isoprene during the summer months. Mixing ratios at Flax
Pond are lower during the winter, but comparable to those observed at ASRC during the same
season (~ 100-150 ppt). Furthermore, the variability is predominantly driven by the isoprene
interference. Figures 9 and S8 demonstrate that interferences will vary spatially between heavily
urban and biogenic-dominated regions. In addition, outside of the summer months, isoprene is
unlikely to be a major contributor to m/z 69 in both regions.
The contribution of the isoprene interference to m/z 69 in New York City is comparable to the
ground level measurements in Las Vegas, Los Angeles, and Detroit (0.25–0.5 ppb), which
demonstrates that isoprene measurements by PTR-MS are likely to be significantly impacted
across most, if not all, urban regions. Identifying anthropogenic or nighttime sources of isoprene
by PTR-MS will be difficult if not confirmed unambiguously by GC-PTR-ToF-MS or separately
by GC-MS.



### 3.2. Oxygenated VOCs

#### 3.2.1. Characterizing interferences to oxygenated VOCs using GC-PTR-ToF-MS

Small oxygenated VOCs are an important contributor to the reactivity and ozone produced in urban areas. Alcohols, ketones, and small aldehydes ($< C_4$) may be emitted to the atmosphere from mobile sources, VCPs, cooking activities, and other sources (Klein et al., 2016;McDonald et al., 2018), but are also formed as secondary products of atmospheric chemistry. Some studies have reported that certain alcohols, such as ethanol, may ionize to product products that overlap with proton-transfer products of other important oxygenates, such as acetaldehyde (Buhr et al., 2002;Pagonis et al., 2019). Previous intercomparisons have shown that acetaldehyde is one example of oxygenated VOCs where there may be large disagreements between PTR-ToF-MS and GC-MS (Yuan et al., 2016).

Figure 10 shows the GC-PTR-ToF-MS chromatogram collected on the Las Vegas Strip for proton-transfer products typically assigned to oxygenated VOCs. Here, we present small oxygenates typically reported in ambient data sets that are subject to fragmentation or interferences, including methanol, acetaldehyde, ethanol, and $C_4$-carbonyls, which represent the sum of methacrolein (MACR), methyl vinyl ketone (MVK), and crotonaldehyde.

First, GC-PTR-ToF-MS data show that no other species elute through the GC column to yield a significant interference to methanol and ethanol. The signal at m/z 59 ($C_3H_6OH^+$) is also observed to result entirely from acetone + propanal (not shown). This is consistent with previous studies that show good agreement between GC-MS and PTR-ToF-MS (e.g., Warneke et al., 2003).

Crotonaldehyde is a major fraction of the $C_4$-carbonyls observed on the Las Vegas Strip. Typically, MVK and MACR are treated as the dominant isomers to the $C_4$-carbonyl product (m/z 71, $C_4H_6OH^+$), since these are secondary products from isoprene oxidation and are expected to be present at high mixing ratios (Yuan et al., 2017). Crotonaldehyde is observed to be a major contributor to m/z 71 in biomass burning emissions (Koss et al., 2018), but its presence on the Las Vegas Strip likely points to other important aldehyde sources, such as cooking. The higher fraction of crotonaldehyde reflects that isoprene mixing ratios are lower in Las Vegas than other cities (Fig. 5) and that cooking is an important source of VOCs along the Las Vegas Strip (Fig. 2). Xu et al. (2022) showed that measurements of $C_4$-carbonyls by the NOAA PTR-ToF-MS, ammonium-adduct chemical ionization mass spectrometer ($NH_4$-CIMS), and NOAA GC-MS agreed in the daytime during RECAP-CA when MVK and MACR were high, but disagreed at night when isoprene products were low and crotonaldehyde mixing ratios were likely elevated. Additional interferences at m/z 71 could result from decomposition of ISOPOOH on inlet surfaces (Rivera-Rios et al., 2014).



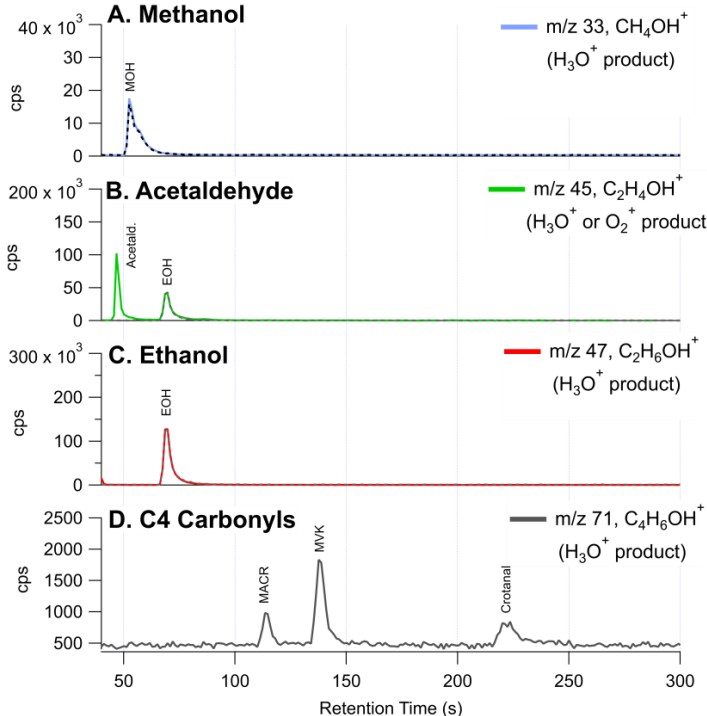

**Figure 10.** GC-PTR-ToF-MS chromatograms from the Las Vegas Strip showing the contributions of isomers and fragments to ions typically assigned to small oxygenates. The labels highlight the traditionally assigned isomers for (a) methanol, (a) acetaldehyde, (c) ethanol, and (d) $C_4$-carbonyls including methacrolein (MACR), methyl vinyl ketone (MVK), and crotonaldehyde.

### 3.2.2. Interferences to m/z 45 from ethanol reactions with $O_2^+$

The most significant interference for the small oxygenated VOCs observed by GC-PTR-ToF-MS is associated with the ionization of ethanol to produce signal at the mass typically assigned to acetaldehyde (m/z 45, $C_2H_5O^+$). Ethanol has been shown by Inomata and Tanimoto (2009) to produce fragments at m/z 19 ($H_3O^+$), m/z 31 ($CH_3O^+$), and m/z 29 ($C_2H_5^+$). Buhr et al. (2002) identified m/z 45 as product and found that it correlated to the ethanol proton transfer product (m/z 47, $C_2H_6OH^+$) with a ratio of 0.22. The likely pathway for the formation of m/z 45 is by ethanol reactions with $O_2^+$, which has been identified by Spanel and Smith (1997) as the dominant $O_2^+$ product using selective ion flow tube (SIFT) mass spectrometry. The NOAA Vocus PTR-ToF-MS observes a ratio that is higher than that determined by Buhr et al. (2002) (~0.38), although the distribution of total fragmentation (i.e., the sum of all ethanol fragments relative to m/z 47) appears similar (Fig. S1).

Figure 11 shows the temporal behavior of m/z 45 and m/z 47 (ethanol) during the nighttime drive on 30 July. Figure 11a shows the mixing ratio of m/z 45 assuming the sensitivity of acetaldehyde and Fig. 11b shows the mixing ratio of ethanol. Figure 11c shows a scatter plot of the signal at m/z 45 vs. that of m/z 47 for the entire mobile laboratory dataset. First, ethanol and m/z 45 are correlated when ethanol mixing ratios are high (Fig. 11a, b). Ethanol on the Las Vegas Strip





reached mixing ratios of 1.5 ppm and corresponding increases in m/z 45 were observed that point
towards a contribution from ethanol. Figure 11c shows that a subset of the m/z 45 signal measured
throughout the Las Vegas dataset exhibit a ratio to ethanol that agrees with the fragmentation ratio
observed from GC-PTR-ToF-MS measurements. These observations point towards a broader
impact of ethanol on m/z 45 throughout the Las Vegas region.

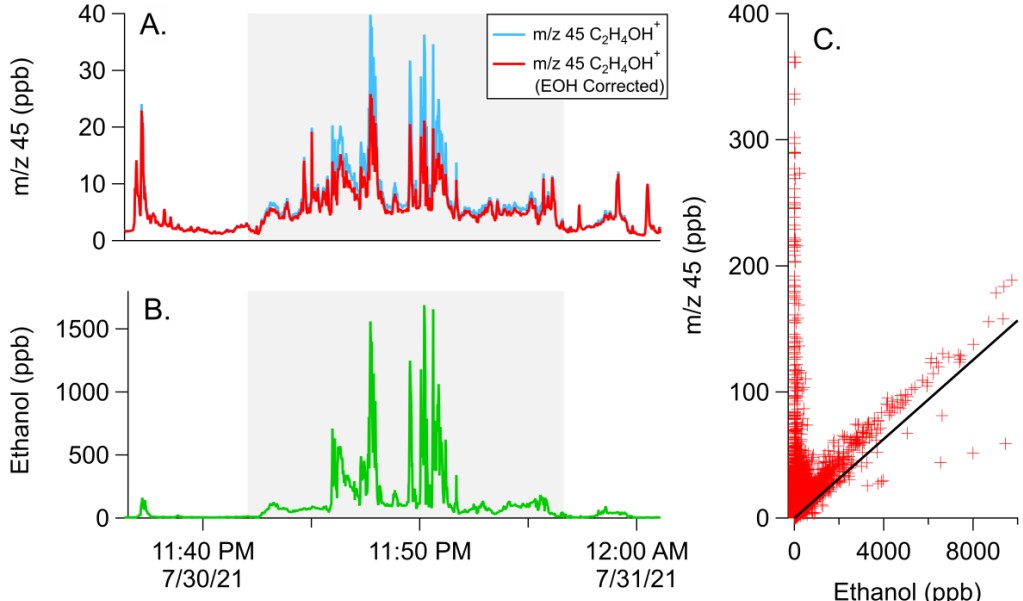

**Figure 11.** Demonstration of impacts of ethanol on mobile drive data in downtown Las Vegas during the evening
drive on 30 July. (a) Time series of the signal at m/z 45 ($C_2H_4OH^+$) with and without the subtraction of the ethanol
interference. (b) Time series of ethanol (m/z 47, $C_2H_6OH^+$). The shaded regions show when the mobile laboratory was
sampling along the Las Vegas strip. (c) Correlation plot of mobile drive data for the entire Las Vegas dataset. The
solid line shows the fragmentation ratio of m/z 45 to m/z 47 for ethanol, as derived from the GC-PTR-ToF-MS data
(Fig. 10).

### 3.2.3. Corrections to m/z 45 measured in Las Vegas

The extent to which ethanol contributed to the signal at m/z 45 can be determined by correction
techniques. Figure 11a shows the m/z 45 signal with the contribution from ethanol subtracted
following:

$$\text{m/z } 45 \text{ }_{Corrected} = S_{45} - S_{47} \cdot f_{45/47} \quad \text{(Equation 2)}$$

$S_{45}$ is the signal from m/z 45, $S_{47}$ is the signal of ethanol, and $f_{45/47}$ is the ratio determined by GC-
PTR-ToF-MS. Generally, the ethanol-corrected data on m/z 45 show that ethanol contributed
~40% to the signal on the Las Vegas Strip. Outside of this region, ethanol ionization has a modest
impact on m/z 45. Over the average mobile laboratory dataset, ethanol may have contributed as
much as 5% to the total signal at m/z 45. Similar contributions are estimated for the ground site
data collected during ground sampling at Caltech. Consequently, ethanol reactions with $O_2^+$ may





only be an important contributor to m/z 45 in highly-concentrated ethanol plumes, which may be
encountered during mobile sampling or upon aircraft encounters with point sources. This ratio may
also be affected by humidity, which changes the distribution of $O_2^+/H_3O^+$ in drift tubes operated
at low water mixing ratio.

The GC-PTR-ToF-MS provides some insights into the interferences of oxygenates, but there are
limits to the extent to which oxygenates elute through a DB-624 or other GC columns with similar
polarity. Interferences towards oxygenated masses may be an important focus for future work, as
recent studies have pointed towards the increasing fraction of oxygenated VOCs observed in urban
air (Karl et al., 2018;Xu et al., 2022;Khare et al., 2022) and instrumentation capable of measuring
unfragmented oxygenates are becoming more common (e.g., Khare et al., 2022;Xu et al.,
2022;Riva et al., 2019). Intercomparisons with GC-MS measurements employing polar columns,
or with mass spectrometers employing softer ionization chemistry (e.g., iodide or $NH_4^+$ adduct
mass spectrometers) may help to better characterize the response and selectivity of PTR-ToF-MS
to oxygenates.

### 3.3. Aromatic VOCs

#### 3.3.1. Known interferences to aromatic masses

PTR-ToF-MS is well suited to measure ambient mixing ratios of $C_6$–$C_9$ aromatics; however, it is
known that alkyl aromatics (e.g., ethylbenzene and ethyltoluene isomers) and aromatic
monoterpenes and monoterpenoids (e.g., cymene and fenchone, Kari et al., 2018;Tani, 2013)
fragment and contribute to the signals typically attributed to benzene (m/z 79, $C_6H_6H^+$) and toluene
(m/z 93, $C_7H_8H^+$) (Pagonis et al., 2019;Yuan et al., 2017). The abundance and distribution of
aromatics depends on the relative mix of VOC emissions from petrochemical sources, including
fossil fuels, solvents emitted from VCPs (e.g. paints and architectural coatings), and asphalt paving
(Gkatzelis et al., 2021a;Gkatzelis et al., 2021b;Khare et al., 2020;Stockwell et al., 2021). Higher
aromatics, such as xylenes and ethylbenzene, are prevalent in both fossil fuel and VCP emissions,
whereas benzene is restricted in consumer products and is therefore almost entirely associated with
fossil fuels (McDonald et al., 2018). Consequently, PTR-ToF-MS measurements in urban regions
with significant solvent emissions may exhibit a greater degree of interference on benzene and
toluene than regions with greater fossil fuel usage.

#### 3.3.2. Characterizing interferences to benzene (m/z 79) using GC-PTR-ToF-MS

Figure 12 shows GC-PTR-ToF-MS measurements of key aromatic compounds measured in
downtown Las Vegas where both fossil fuels and VCP emissions were prevalent. Each panel is
labeled by the typical compound assignment and shows chromatograms of the corresponding
proton-transfer product. In general, chromatograms show that $C_9$- and $C_8$-aromatics are the
expected key contributors to the signals at m/z 121 ($C_9H_{12}H^+$) and m/z 107 ($C_8H_{10}H^+$),
respectively. This is consistent with previously observed PTR-ToF-MS behavior (Yuan et al.,
2017), and shows that urban measurements at these masses continue to be reliably assigned to
simple alkyl aromatics.

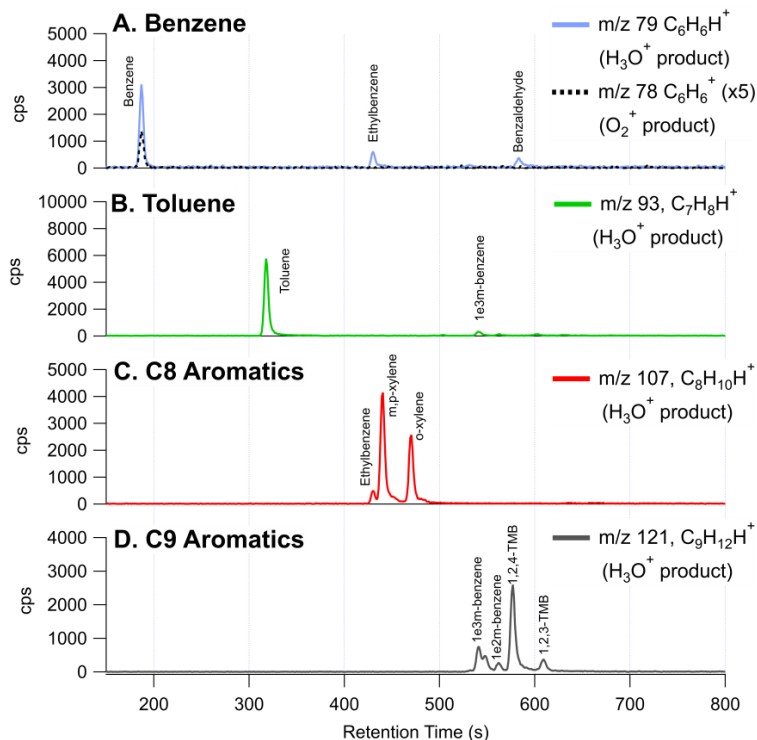

**Figure 12.** GC-PTR-ToF-MS chromatogram from downtown Las Vegas showing the contributions of isomers and
fragments to ions typically assigned to $C_6$–$C_9$ aromatics. The labels highlight the traditionally assigned isomers for (a)
benzene, (b) toluene, (c) $C_8$-aromatics including *o,m,p*-xylene + ethylbenzene, and (d) $C_9$-aromatics including
ethyltoluene isomers + trimethylbenzene isomers.

In contrast, the masses typically assigned to benzene (m/z 79, $C_6H_6H^+$) and toluene (m/z 93,
$C_7H_8H^+$) show greater contributions from the fragmentation of alkyl aromatics. At m/z 93, most
of the signal is attributed to toluene and a small fraction ($< 5\%$) results from the fragmentation of
1-ethyl-3-methylbenzene. At m/z 79, ~ 80% of the signal results from the proton-transfer product
of benzene and the remainder from the fragmentation of ethylbenzene and benzaldehyde. Previous
work has shown contributions of ethylbenzene to m/z 79 in urban air (Inomata et al., 2010),
whereas contributions from benzaldehyde are not well studied. Benzaldehyde may result from
VCPs, cooking, motor vehicle emissions, biomass burning, or secondary production (Gkatzelis et
al., 2021a;Koss et al., 2018;McDonald et al., 2018).

**3.3.3. Corrections to ground site observations of m/z 79 in Los Angeles and Las Vegas**

The interferences at m/z 79 are significant and present a challenge for reliably quantifying benzene
in Las Vegas and other urban regions. To quantify this interference, Fig. 13 highlights benzene
measurements from the Jerome Mack and Caltech ground sites. Figure 13a, b show corrected and
uncorrected benzene at m/z 79 can be attributed to benzene calculated from two methods:

$$\text{m/z } 79_{\text{Corrected}} = S_{\text{C6H6H+}} - S_{\text{C7H6OH+}} \cdot f_{79/\text{Benzald}} - S_{\text{C8H10H+}} \cdot f_{79/\text{Ethylbenzene}} \quad \text{(Method 1)}$$



$$m/z\ 79_{Corrected}\ =\ S_{C6H6+}\quad (\text{Method 2})$$

Where $S_{C6H6H+}$ is the signal of $C_6H_6H^+$, $S_{C7H6OH+}$ is the signal attributable to benzaldehyde, $S_{C8H10H+}$ is the signal attributable to ethylbenzene, and $S_{C6H6+}$ is the signal attributed to the benzene charge-transfer product at m/z 78. $f_{79/Benzald}$ and $f_{79/Ethylbenzene}$ are the fragmentation patterns of benzaldehyde ($f_{79/Benzald} = C_6H_6H^+/C_7H_6OH^+$) and ethylbenzene ($f_{79/Ethylbenzene} = C_6H_6H^+/C_8H_{10}H^+$) as determined by GC-PTR-ToF-MS chromatograms. Method 1 corrects for benzene by subtracting the contributions of benzaldehyde and ethylbenzene to the signal at $C_6H_6H^+$. GC-PTR-ToF-MS measurements show that benzaldehyde is the primary contributor to $S_{C7H6OH+}$, whereas ethylbenzene is one of four isomers that contributes to $S_{C8H10H+}$. We use the GC-PTR-ToF-MS measurements at the Jerome Mack ground site and find that ethylbenzene contributes ~12.5% of the total $C_8$-aromatic signal. Method 2 simply estimates benzene mixing ratios based on calibrations applied to the charge-transfer product at m/z 78 ($C_6H_6^+$). This mass has no discernible interference from other VOCs in the GC-PTR-ToF-MS data (Fig. 12) and is detected with sufficient sensitivity to reliably quantify benzene (~180 cps ppb$^{-1}$). We note that Method 1 requires regular quantification of $C_8$-aromatic distributions by GC in order to account for ethylbenzene fragmentation, whereas Method 2 relies only on measurements of the $O_2^+$ charge-transfer product. We note that Method 2 may present limitations if other species are present that fragment to produce the $O_2^+$ product. Deployment of GC-PTR-ToF-MS in other cities may help to determine whether the charge-transfer product is unambiguously linked to benzene.

On average, the interferences from ethylbenzene and benzaldehyde constitute ~ 38% of the signal at m/z 79 detected at Jerome Mack, and ~ 3% of signal detected at Caltech (pie charts, Fig 13). We speciate the interference at Jerome Mack using GC-PTR-ToF-MS and find that the majority of the interference is associated with fragmentation of ethylbenzene (31% of total signal) with a small contribution from benzaldehyde (7% of total signal). Ethylbenzene was likely emitted from a local source due to a cabinet-making shop upwind of the ground site. The two methods for correcting benzene agree well for Jerome Mack data, which confirms that ethylbenzene and benzaldehyde are the primary contributors to the benzene interferences. We note that we only use Method 2 for data collected at Caltech since GC-PTR-ToF-MS measurements were unavailable during this period of the deployment.



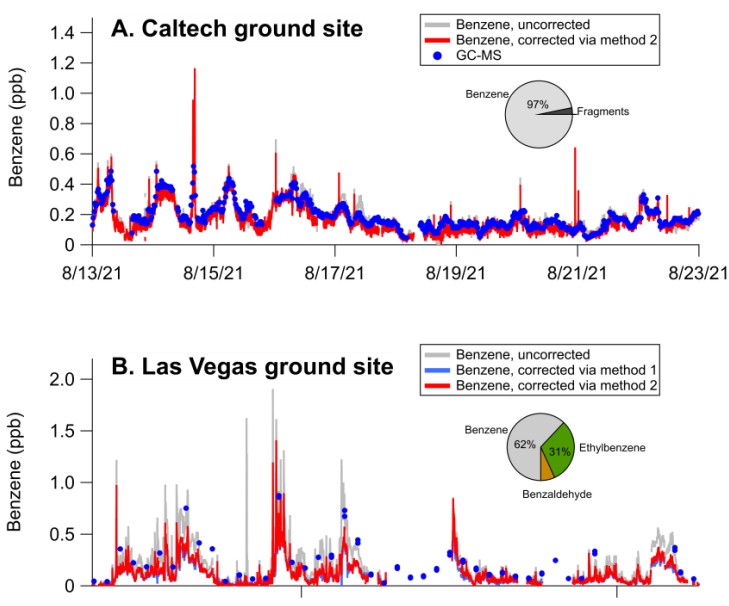

**Figure 13.** Impact of fragmentation on the signal at m/z 79 ($C_6H_6H^+$) and corresponding benzene mixing ratios measured at (a) Caltech and (b) the Jerome Mack ground site in Las Vegas. The corrections using the two methods are shown compared to uncorrected data. The pie charts show the average contribution of benzene, ethylbenzene, and benzaldehyde to the signal at m/z 79 ($C_6H_6H^+$).

### 3.3.4. Corrections to observations of m/z 79 in Detroit

The significant interferences to m/z 79 observed in Las Vegas are also observed in PTR-ToF-MS data collected downwind of Detroit during the MOOSE campaign. Figure 14 shows four days of mobile laboratory sampling with the Aerodyne PTR-ToF-MS and GC-MS instrumentation. Similar to Fig. 13, corrected via Method 2 and uncorrected m/z 79 measurements are shown (Fig. 14a). Figure 14b shows the histograms of PTR-ToF-MS measurements alongside those from the GC-MS during the entire campaign. The pie chart shows the average fraction of m/z 79 attributed to benzene vs. the fraction associated with fragments. Benzaldehyde and ethylbenzene contributions are not separated since GC-PTR-ToF-MS measurements were unavailable.

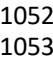
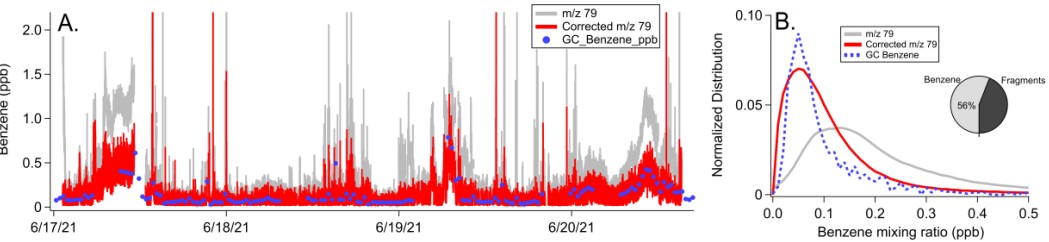



**Figure 14.** (a) Time series of GC-MS samples of benzene and PTR-ToF-MS mixing ratios of m/z 79 ($C_6H_6H^+$) and corrected m/z 79 reported during sampling by the Aerodyne Mobile Laboratory downwind of Detroit, MI. (b) Histograms showing the distribution of m/z 79 measured by the Aerodyne PTR-ToF-MS, corrected m/z 79 mixing ratios calculated using Method 2, and benzene mixing ratios measured by the GC-MS. The pie chart highlights the fraction of the m/z 79 associated with benzene vs. the fraction associated with fragments.

Similar to Las Vegas, fragmentation of higher aromatic species plays an important role in determining the benzene signal in m/z 79. The distribution of uncorrected m/z 79 shows a peak around 0.12 ppb and a broad tail biased towards higher mixing ratios. The GC-MS measures a distribution of benzene with a maximum at 0.05 ppb, and a much lower frequency of higher mixing ratios. When the PTR-ToF-MS data are calibrated using the benzene charge-transfer product (Method 2), corrected m/z 79 mixing ratios show a better agreement with GC-MS measurements. The distribution of corrected m/z 79 is wider than that reported by GC-MS, which may reflect the faster sampling of the PTR-ToF-MS and more frequent observations of concentrated aromatic plumes (Fig. 14a). Over the entire sampling period, the average distribution of m/z 79 shows that benzene accounts for ~ 56% of the total signal. This is consistent with the observations in Las Vegas (interference ~ 62% of the signal), indicating that m/z 79 in both datasets were influenced by solvent emissions to a significant extent.

## 4. Conclusions

Urban VOCs significantly contribute to the degradation of air quality, and PTR-ToF-MS provides important constraints on the emissions and chemical transformation of many gas-phase organics. Advances in PTR-ToF-MS sensitivity and detection provide opportunities to identify, characterize, and revisit measurement interferences to commonly reported VOCs (Yuan et al., 2017). Here, we find that long-chain aldehydes, along with previously identified cycloalkanes, are important contributors to the signal of m/z 69 typically associated with isoprene in many urban areas. The fragmentation of these molecules can be larger than the signal associated with the proton-transfer product from isoprene, depending on the mixture of anthropogenic and biogenic VOCs, time of day, and season. We find that interferences at ground level in Los Angeles (large isoprene emissions, large anthropogenic emissions) are highest at night and constitute ~10% of the signal observed during the day. Interferences are a higher fraction of m/z 69 at altitude (> 50%) and are observed to be widespread throughout the Los Angeles Basin. In Las Vegas (low isoprene emissions, large anthropogenic emissions), interferences dominate the signal at m/z 69 throughout the day and night. These interferences are observed in other cities (e.g., Detroit and New York City), depend on season, and are common among drift tube designs operated at similar E/N.

Other PTR-ToF-MS masses also exhibit interferences, including those typically assigned to oxygenates and aromatic VOCs. Fragmentation from ethanol impacts measurements of acetaldehyde on m/z 45, though this interference is only significant in regions with large ethanol emissions. PTR-ToF-MS measurements of benzene using m/z 79 exhibit significant interferences from the fragmentation of ethylbenzene and benzaldehyde. In Las Vegas and Detroit, fragmentation impacts m/z 79 mixing ratios by as much as 40%. The growing contribution of interferences to aromatics and oxygenates may reflect the changing mix of urban VOC emissions from one dominated by mobile sources to one dominated by solvents (Gkatzelis et al., 2021b;McDonald et al., 2018). In the case of benzene, other ions such as the charge-transfer product (m/z 78, $C_6H_6^+$) can be used to quantify benzene without significant influence from



fragmentation from higher carbon VOCs. As instrument sensitivity improves, there may be other
ions that can be used to improve the quantification of additional VOCs.
Corrections to these interferences are feasible, though it is unlikely that a universal correction
factor is sufficient to resolve instrument discrepancies across datasets. Instrument responses, as
well as changes to the VOC mixture in different regions, require that detailed characterization be
performed on a dataset-by-dataset basis. GC-MS measurements provide an opportunity to compare
against PTR-ToF-MS measurements for a wide-variety of key VOCs, including isoprene, small
oxygenates, and aromatics. Likewise, information about fragmentation and instrument-specific
responses to reactive hydrocarbons can be determined using GC-PTR-ToF-MS. For species such
as oxygenates, intercomparisons against other mass spectrometers using softer ionization (e.g.,
iodide or ammonium-adduct CIMS) or use of GC pre-separation using polar columns may yield
valuable information about instrument artifacts.
**Data Availability**
Data for SUNVEx and RE-CAP are available at the NOAA CSL data repository
(https://csl.noaa.gov/projects/sunvex/). Data for FIREX-AQ, MOOSE, and LISTOS are available
at the NASA data repository (https://www-air.larc.nasa.gov/missions.htm).
**Author Contribution**
MMC, CES, XL, JBG, AL, CW, EYP, CA, EFK, and AHG conducted measurements during
SUNVEx and RE-CAP. MMC, GIG, CW, JBG, AL, AW, FP, DB, RH, and ECA conducted
measurements during FIREX-AQ. MSC, BML, FM, and MC conducted measurements during
MOOSE. JM, CC, and JEM conducted measurements during LISTOS. MMC and CW wrote the
paper with contributions from all authors.
**Competing Interests**
EAC is a co-editor of Atmospheric Measurement Techniques. The peer-review process was guided
by an independent editor and the authors also have no other competing interests to declare.
**Acknowledgements**

MMC, CES, XL, JBG, and CW acknowledge funding from Clark County, NV (contract number
20-022001) and the California Air Resources Board (contract number 20RD002). JM, CC, and JM
acknowledge funding from the Northeast States for Coordinated Air Use Management
(NESCAUM) through a contract with the New York State Energy Research and Development
Authority (NYSERDA) (agreement number 101132). AHG, EYP, EFK, and CA acknowledge
funding from the California Air Resources Board (contract number 20RD003, 20AQP012), NOAA
Climate Program Office's Atmospheric Chemistry, Carbon Cycle, and Climate program (contract
number NA22OAR4310540 [UCB]/ NA22OAR4310541 [AD]). the Office of Naval Research
Defense University Research Instrumentation Program (grant number N00014-19-1-2108), and
EPA-STAR (grant number 84001001). EYP was supported by a Feodor Lynen Fellowship of the
Alexander von Humboldt Foundation.



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
