# Peer review of "Identifying and correcting interferences to PTR-ToF-1"

_EGUsphere, 2023_

## Author Comment (AC1)

We thank the reviewers for their careful reviews and helpful comments to our manuscript. We believe these comments have helped to improve the organization and presentation of the material. Below is a point-by-point response to reviewer comments.

**Response to Reviewer 1**

*That being said, the work presented would be strengthened if an estimated uncertainty was presented for each technique employed for the target species discussed (i.e. JCGM 100: 2008 Evaluation of Measurement Data – Guide to the Expression of Uncertainty in Measurement, BIPM, Sevres, France, 2008). This would allow us to see the progress in addressing uncertainties in commonly applied VOC measurement approaches that may currently be poorly understood/quantified.*

We thank the reviewer for this comment. The uncertainties of PTR-ToF-MS and GC-MS measurements of calibrated species are typically determined by uncertainties in the delivery of gas standards. This variability is determined by uncertainties of the mixing ratios in gas standards, uncertainties in the delivery of gas standards by mass flow controllers, and day-to-day variability in instrument responses to those mixtures. Uncertainties are often reported based on campaign statistics in the instrument response to calibration standards. There are also uncertainties reported for instrument intercomparison studies. For PTR-ToF-MS, fragmentation increase measurement uncertainty, as shown by this study and elsewhere

We now provide uncertainties for calibrated species in the description of each instrument.

*The authors could also consider demonstrating the impact of these corrections by reporting the difference in estimated ozone or SOA formation potentials determined from corrected and uncorrected data.*

Thank you for this suggestion. We agree that this is important, especially for non-attainment regions such as the Los Angeles Basin or New York City where air quality exceedances are frequent and biogenic emissions are an important contributor to ozone pollution. Quantifying these impacts is challenging since measurements only provide local information about VOC mixing ratios, whereas pollution in the LA Basin results from a heterogeneous mixture of emission sources. We do believe that studies relying on PTR-ToF-MS measurements to constrain emissions inventories (e.g., BEIS, MEGAN) need to consider these corrections in order to properly estimate biogenic contributions to VOC reactivity, ozone formation, and SOA.

We have provided a brief section at the end of Section 3 and in the conclusions that describe the implications of the interferences on air quality models.

*Include info on type of regression analysis used (LLS, ODR, RMA). If LLS has been used consider other methods that account for random measurement error on both axis.*

This is a good suggestion – we initially used LLS, but have transitioned to use an ODR fit. The slopes for the isoprene correction changed by ~3% and did not substantially change the results shown for the Las Vegas and Los Angeles data. We note the use of an ODR fit in the description of Eq. 1.

*If sufficient concentrations are present, consider applying the Method 2 benzene correction approach presented in Section 3.3.3 to isoprene using m/z 68? This could be of relevance for stand-alone PTR-ToF-MS measurements.*

Unfortunately, our GC-PTR-ToF-MS measurements showed that m/z 68 was influenced by the fragmentation of larger VOCs. This was different for benzene, which did not exhibit significant interferences. Consequently, we are not confident that m/z 68 is a good measure of isoprene in urban areas.

*Spaces needed between references presented in brackets in text*

This has been corrected.

Suggest consistent colouring in figures. In Figures 3, 4, 6, 9 red is used to show interference whereas in Figures 5,7,8,11,13,14 red is used for the corrected concentration.

Thank you for catching this, the colors have been updated to be consistent.

Include actual value in brackets where slopes and interference ratios are reported in the text.

This has been updated to reflect measured ratios in brackets.

**Response to Reviewer 2**

To make this resource most effective for the future reader, the text could be organized by *instrument* rather than by campaign. As a resource, I am much more interested in comparing my own results with the same instrument reported in the text. Then, I can see how the environment (urban vs rural for example) could impact the fragmentation. This proposed changed would involve some copy-pasting to rearrange the text so that each instrument is described in detail, and then following this description; its deployment is listed. Some field campaigns would then be reiterated, but the figures wouldn't need to be updated. My comment here is really coming from the perspective of a PTR user and using this manuscript as a reference.

We thank the reviewer for their careful review of our manuscript and special attention to improve the organization for use as a reference to PTR users. We believe that the suggestions made by the reviewer have helped to streamline the discussion and will enable the paper to be more easily used by the community.

As noted by the reviewer, our manuscript is ordered by campaign. This choice was because our primary focus is to describe the causes of interferences, the situations where interferences might arise, and methods to identify / correct for these interferences. As a result, we place less focus on instrument-to-instrument variability in these responses, and more focus on the mechanics for correcting data.

To better serve the PTR community while still maintaining this purpose, we have increased the number of subsections and changed the section headers to identify the instruments for which interferences are being described. Our previous manuscript described the result of each instrument in distinct sections, but as the reviewer noted, this was not directly called out in the manuscript. We believe this makes it easier for the reader to find sections where the measurements of comparable instruments are located. We have also changed Table 1 to be more PTR-focused. We believe this table provides a better comparison between instruments and highlights regions where the performance of multiple PTR-MS instruments can be compared (e.g., Los Angeles). Finally, we have moved the description of the GC-MS instruments to the supplement to ease the text in the manuscript, which we feel places higher focus on the description of the PTR instruments.

**Scientific comments:**

Generally, it seems that m/z 69 and isoprene are used interchangeably in this study (comparing Figure 5 and 7), but as the authors point out, they represent different values in this study: (1) m/z ratio obtained by the PTR or (2) parent ion of isoprene. Points to address to help bring consistency:

L 737 "...m/z 69 varied between 200–1200 ppt…" It isn't accurate to report a mixing ratio for a m/z ratio, because one can only calibrate for a specific VOC. So mixing ratios should be used when mentioning isoprene specifically.

We agree with the assessment that mixing ratios of a mass are not completely accurate. However, our intention in using "mixing ratios" when describing m/z 69 is to calibrate the reader to the extent that fragmentation may lead to an overestimation in isoprene mixing ratios. To clarify our meaning, we now include a short paragraph at the beginning of Section 3 (reproduced below). We also note that our analysis is specific to high-resolution PTR-MS systems since low-resolution systems measure ions that are isobaric to $C_5H_8H^+$.

"A key focus of this discussion is the impact of fragmentation on PTR-ToF-MS observations of the $C_5H_8H^+$ ion at m/z 69.070 (henceforth referred to as "m/z 69"). This ion is typically assigned to isoprene and mixing ratios often determined based on measurements of isoprene standards; however, the raw signals at this mass may result from contributions of other compounds. Throughout the discussion, we report m/z 69 mixing ratios calculated using PTR-ToF-MS sensitivities towards isoprene in order to demonstrate the extent to which interferences bias estimations in isoprene mixing ratios. We note that this analysis only reflects interferences attributable to high-resolution PTR-MS systems. Quadrupole or compact time-of-flight mass spectrometers measure additional ions that are isobaric to $C_5H_8H^+$ (e.g., furan, $C_4H_4OH^+$ at m/z 69.034) and further corrections are needed to these data to resolve isoprene mixing ratios."

Would it then be worth reporting counts per second throughout the discussion on m/z 69?

We hope that the clarification above provides clarity and consistency to our discussion.

Some instruments have unit mass resolution (m/z 69) and some have high mass resolution (m/z 69.070). It would be worth being consistent throughout the text. (L 491-493).

We agree and now include a statement that our analysis is primarily focused on high mass resolution data (see above).

Table 1 could also be revised to include m/z ratios

We agree and now include m/z ratios in the table.

L 933-935, 1105 -1106: Great discussion on how instrument discrepancies could affect the correction factor. Thus, authors could consider discussing some of the statements (for example, RH dependencies, Charge transfer ionizations) in light of instrument specificity. These factors will vary for different instruments.

Thank you for this suggestion – we agree and have expanded at these parts of the manuscript to describe how differences in instrument design could affect these factors.

**Organization/formatting comments:**

To help make this resource even more accessful to the future reviewer and reader, I can encourage the authors to consider:

adding more sub and subsubsections to their text. As it stands, there are sections that go on for ~5 pages (section 3.1).

We have added subsections to help streamline the discussion and help readers to identify parts of the manuscript where specific measurements are described.

adding to table 1 the years of the campaign and whether the instrument is from Ionicon or Tofwerk/Aerodyne or custom-built or another source. Since PTR instruments have differences in their drift tubes, highlighting this difference is necessary for the future user. This table would be an excellent summary and resource for the future reader, and could be as large as taking up one whole landscape page with all the details of each instrument. What I'm suggesting here is having a summary table for quick references on similariries and differences of instrument operation (arguably more important than the campaign results).

We have revamped the table to focus on instrument details, as suggested by the reviewer. These details are now included in Table 1.

presenting the a merged methods and results sections, perhaps per instrument and then applied to a campaign.

We feel that a merged methods / results section would not fit with our focus to describe the mechanics of identifying and correcting interference. We would prefer to keep these separated, but have adopted the reviewers suggestions to help clearly identify the places in the manuscript where the results of individual instruments are described.

Figure 13 and 14, the pie charts are concise demonstrations of the importance of applying the proposed corrections, maybe those could be added to the previous figures as well where applicable?

We thank the reviewer, and agree that the pie charts are helpful to visualize the importance of the corrections. The pie charts in 13 and 14 represent fractions that are consistent across most measurements at these sampling locations. This is not the case for isoprene or acetaldehyde which have larger temporal and spatial variations. We feel that an average pie chart for these species would not show the importance of the correction, especially at night when m/z 69 is almost entirely associated with fragmentation.

**Some technical comments:**

L50-53: Would be worth defining secondary ionization and help clarify how the drift tube designs make an important difference in how much charge transfer can occur vs proton-transfer.

This point can be reiterated on lines 100-102 as this process is quite different between Ionicon vs Tofwerk/Aerodyne instruments.

Great point – we have added these details in the discussion. We also provide some details about the differences in the Ionicon and Tofwerk instruments and how this can impact ionization.

L79,84, 90: the loss of OH2 through a "dehydration" process would involve a change in oxidation state at the carbon of an organic molecule. As a chemist, the process of dehydration is significantly different than the process of fragmentation. Do they authors want to clarify their point further? (also an aldehyde can be written as (C(O)H, not with a doublebonded H)

This is a good point – we've changed this description to read "neutral loss of water" as opposed to dehydration to be consistent with previous papers (e.g., Pagonis et al.). We have updated this throughout the text where "dehydration" is implied.

L81-82: would be worth adding a reference to support the statement on known fragmentation of alcohols and aldehydes through PTR ionization.

We now reference Buhr et al. here.

L123: VCPs should be defined as it's the first use of this acronym.

We now define VCPs here.

L173: a 10m inlet might also play an important factor in observed signal by PTR. Can the authors comment on the sample line effects?

The sample lines may certainly play a role in instrument responses to VOCs by changing the distribution of VOCs. We have included the following statement to note this effect:

"Long sample lines may alter VOC distributions by acting as a source, sink, or reactive surface for various compounds. These effects may contribute to the variability of interferences discussed here."

L203 " Each flight covered approximately 5000 km of distance across the Los Angeles area,.." According to this website (https://man.fas.org/dod-101/sys/ac/uv-18.htm) the UV-18A Twin Otter has a range of 700 miles. Is it possible that the distance is instead 500 km (or there's a typo?)? Also, who is the vendor of the NCAR TOGA-TOF?

Thank you, this was a typo. It should be 500 km. The TOGA-TOF was custom-built at NCAR, but uses a TOFWERK mass spectrometer.

L 348-349 "The Vocus was operated at a pressure of 2.2 mbar and axial voltage gradient of 600 V, corresponding to an E/N ratio of 125 Td." What T was the reactor at? Including the temperature, pressure and electric field gradient for all instruments to streamline

intercomparability is important for the future reader (that info should also be available for each instrument):

We have added this information to Table 1

L 464-473 This section seems to be a duplicate of L 442-455

Yes, thank you for catching that!

L 635 The Los Angeles or Caltech sites are the same correct (referring to the results in Fig 5)? Could be worth sticking to one name (to facilitate the control find function is a future reader is looking for LA info specifically within this resource)

Yes, these are the same. We will make this specific to Los Angeles throughout the paper. We will also do the same for Jerome Mack (Las Vegas).